# Birth of protein folds and functions in the virome

Jason Nomburg[1,2,3], Erin E. Doherty[3,4], Nathan Price[1,2,3], Daniel Bellieny-Rabelo[3,4], Yong K. Zhu[1,2,3] & Jennifer A. Doudna[1,2,3,4,5,6,7 ✉]

The rapid evolution of viruses generates proteins that are essential for infectivity and replication but with unknown functions, due to extreme sequence divergence[1]. Here, using a database of 67,715 newly predicted protein structures from 4,463 eukaryotic viral species, we found that 62% of viral proteins are structurally distinct and lack homologues in the AlphaFold database[2,3]. Among the remaining 38% of viral proteins, many have non-viral structural analogues that revealed surprising similarities between human pathogens and their eukaryotic hosts. Structural comparisons suggested putative functions for up to 25% of unannotated viral proteins, including those with roles in the evasion of innate immunity. In particular, RNA ligase T-like phosphodiesterases were found to resemble phage-encoded proteins that hydrolyse the host immune-activating cyclic dinucleotides 3',3'- and 2',3'-cyclic GMP-AMP (cGAMP). Experimental analysis showed that RNA ligase T homologues encoded by avian poxviruses similarly hydrolyse cGAMP, showing that RNA ligase T-mediated targeting of cGAMP is an evolutionarily conserved mechanism of immune evasion that is present in both bacteriophage and eukaryotic viruses. Together, the viral protein structural database and analyses presented here afford new opportunities to identify mechanisms of virus–host interactions that are common across the virome.

Viral proteins carry out functions that are critical for infection. Some proteins or their component domains are widely conserved within and across viral families, including between viruses of distinct Baltimore classifications[4] and in viruses that infect different kingdoms of life[5,6]. These include 'viral hallmark genes', such as the jellyroll folds of viral capsid proteins and folds related to RNA- and DNA-directed RNA polymerases[4,7]. However, a major challenge to understanding viral infection mechanisms and evolution is the high percentage of viral proteins with unknown function. Sequence similarity between viral proteins and other viral or non-viral proteins can sometimes suggest protein functions, but the rapid pace of viral evolution and de novo emergence of genes generate many proteins without annotated sequence homologues. This creates a pressing need for alternative approaches to identify protein analogues.

Viral proteins are highly divergent even within the same virus family, limiting the utility of sequence-based similarity searches[8–10] when amino acid identity falls below 30%. By contrast, horizontal gene transfer among viruses and between viruses and cells creates structural relationships that can inform about protein function if they can be detected[3,11,12]. However, viral proteins have limited representation among experimentally determined structures in the Protein Data Bank (PDB) and they are absent from the predicted protein structures in the AlphaFold database[2,13,14].

To address this gap and develop a means to systematically predict viral protein functions, we generated a database of predicted structures from 67,715 proteins encoded by 4,463 species of eukaryotic viruses. We clustered these proteins by sequence and structure, generating 5,770 multi-member and 12,422 singleton clusters. Structural similarity searches greatly expanded the taxonomic diversity of protein clusters, revealing putative protein functions by connecting unannotated viral proteins with annotated analogues. Structural comparisons between viral and non-viral proteins identified potential functions of proteins encoded by human pathogens. In particular, RNA ligase T (LigT)-like phosphodiesterases (PDEs) emerged from this analysis as a widespread class of enzymes that is conserved across the bacterial and eukaryotic virome. Conservation evident within our viral protein structure database, together with enzymatic activities validated in cell-based experiments, reveal an ancient and fundamental role of these proteins in viral anti-immunity pathways.

## The proteome of eukaryotic viruses

To analyse the diversity of protein structures present in eukaryotic viruses, we used ColabFold[15] to predict the structures of 67,715 proteins from eukaryotic viruses included in RefSeq based on viral multisequence alignments (MSAs) (Methods). We then implemented a two-step approach to cluster them, using both sequence-based and structure-based clustering[3] (Fig. 1a). We used MMseqs2[16] to cluster protein sequences to 70% coverage and 20% identity, resulting in 21,913 sequence clusters. Next, we leveraged the alignment speed of Foldseek[17]

[1]Gladstone–UCSF Institute of Data Science and Biotechnology, San Francisco, CA, USA. [2]Department of Molecular and Cell Biology, University of California, Berkeley, Berkeley, CA, USA. [3]Innovative Genomics Institute, University of California, Berkeley, Berkeley, CA, USA. [4]California Institute for Quantitative Biosciences, University of California, Berkeley, Berkeley, CA, USA. [5]Howard Hughes Medical Institute, University of California, Berkeley, Berkeley, CA, USA. [6]Molecular Biophysics and Integrated Bioimaging Division, Lawrence Berkeley National Laboratory, Berkeley, CA, USA. [7]Department of Chemistry, University of California, Berkeley, Berkeley, CA, USA. ✉e-mail: doudna@berkeley.edu

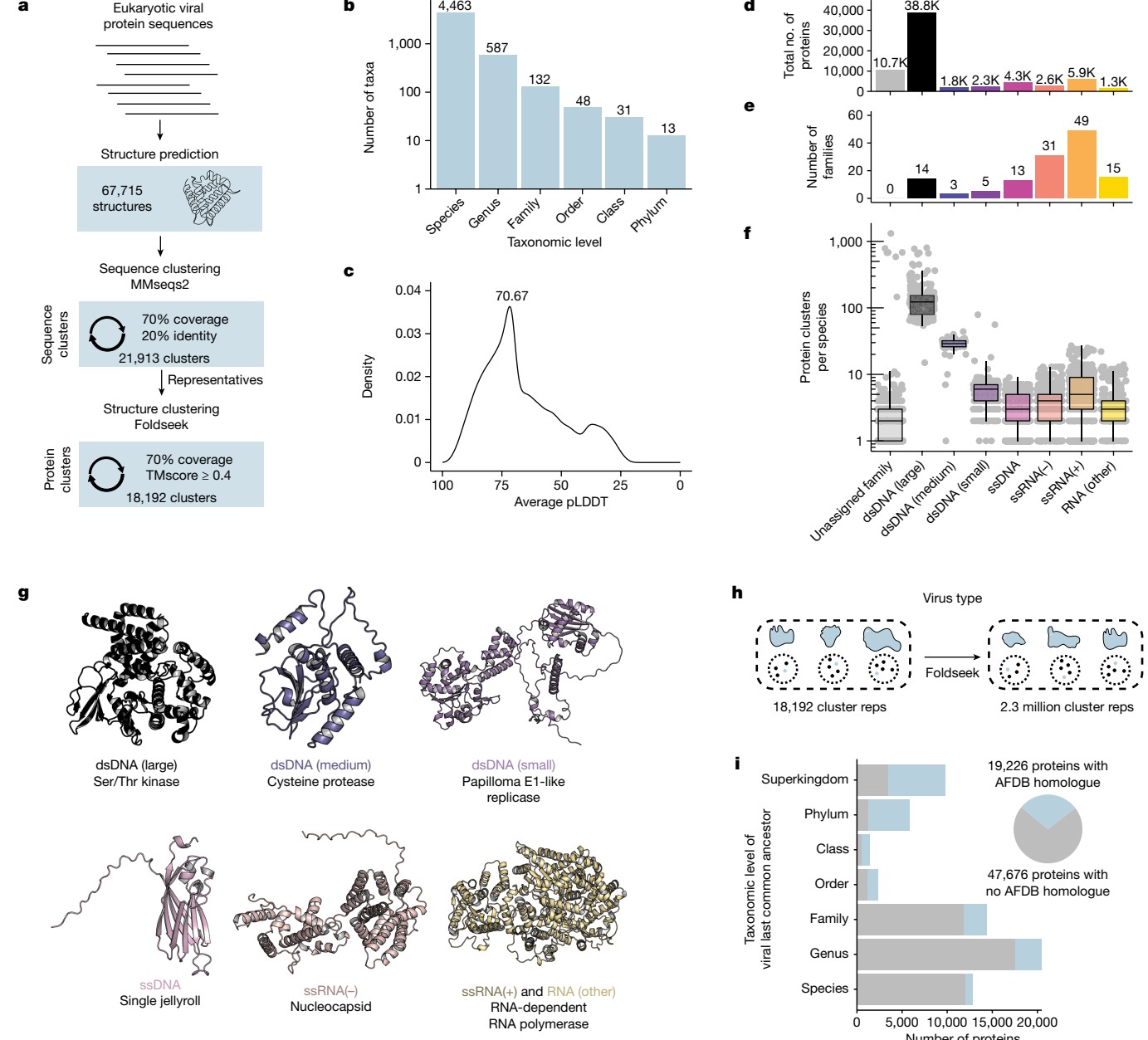

**Fig. 1 | The structural proteome of eukaryotic viruses. a**, Pipeline for protein clustering. Protein sequences from eukaryotic viruses were folded using ColabFold. Protein sequences were clustered to 70% coverage and 20% identity. The predicted structures of the representatives of each cluster were then aligned and clustered together with a requirement of 70% coverage across the structural alignment and a TMscore ≥0.4. This resulted in a final set of 18,192 clusters. **b**, Taxonomic distribution of the dataset. Each column indicates the number of taxa present. **c**, Distribution of the average pLDDT of all structures in the dataset. **d–f**, Viral families were classified by genome type, and the total number of proteins (**d**), viral families (**e**) and protein clusters per species (**f**) are indicated. In box plots, the centre line is the median, box edges delineate 25th and 75th percentiles, and whiskers extend to the highest or lowest point up to 1.5 times the inter-quartile range. **g**, Protein structures representing the protein cluster that is encoded by the highest number of viral families of each genome type. **h**, Foldseek was used to align a single representative protein from each viral protein cluster against 2.3 million clusters generated from the AlphaFold database. **i**, Left, taxonomic level of the last common ancestor of each viral protein cluster was determined. For example, if a protein cluster is encoded by viruses from different orders but the same class, they are placed in the class row. Blue indicates that proteins belong to a cluster with an analogue in the AlphaFold database (AFDB), whereas grey indicates that proteins belong to a cluster without an analogue in the AlphaFold database. Right, pie chart indicating the total number of proteins that belong to clusters whose representatives aligned to the AlphaFold database (blue) or did not align (grey).

to conduct structural alignments between a single representative of each sequence cluster and filtered alignments to keep those with at least 70% alignment coverage, a TMscore of at least 0.4, and an *E*-value lower than 0.001. The resultant structural alignments had a median TMscore of 0.52 (Extended Data Fig. 1a), reflecting robust structural similarity[18]. The 70% alignment coverage threshold enriches clusters for members that are similar across the majority of their protein sequence.

Cumulatively, this resulted in 18,192 protein clusters, of which 12,422 have a single member (Extended Data Fig. 1b). This dataset includes a large diversity of viruses, including 4,463 species from 132 different viral families (Fig. 1b). Clusters are structurally consistent, as implementing DALI[19] to align cluster representatives to each member for clusters with at least 100 members yields a median cluster-average DALI *z*-score of 13.1 (Extended Data Fig. 1c). DALI *z*-scores above 8 indicate

that 2 proteins are likely to be homologous[20]. Proteins in single-member clusters have substantially lower predicted local distance difference test (pLDDT) values than those in non-singleton clusters (Extended Data Fig. 2a), suggesting that structure prediction quality has a major impact on our ability to detect structural similarity. We tested whether MSA generation against a larger reference database has an effect on prediction quality. We found that whereas singletons have a lower average MSA depth that correlates with their lower pLDDT, this alternative MSA generation led to negligible effect on structure prediction quality (Extended Data Fig. 2b–d).

We investigated how well this database represents viral diversity, and if it reconstitutes core viral hallmark genes. We grouped viral families into viral genome types based on the basis of their Baltimore classes with slight modifications—DNA viruses were split into large, medium and small groupings on the basis of their average genome length, whereas RNA viruses without single-stranded positive-sense or negative-sense genomes were grouped into the RNA (other) category. Large double-stranded DNA (dsDNA) viruses have the most protein clusters per species and, despite constituting only 14 of the 132 viral families in the dataset, account for the majority of viral proteins (Fig. 1d,f). As expected, protein cluster count correlates strongly with genome size (Extended Data Fig. 1d). With their larger genomes, dsDNA viruses have the capacity to encode more auxiliary genes without sacrificing genome stability. RNA viruses make up a large fraction of the families present in the dataset, but a smaller fraction of the total proteins (Fig. 1e,f). Structural similarity between viral families with a similar genome type is common, with large dsDNA viruses sharing many protein folds (Extended Data Fig. 1e).

As expected, the predominant protein clusters in the dataset as a whole (Extended Data Fig. 1f) and within each genome type (Fig. 1g) are largely involved in fundamental aspects of the viral life cycle. These include the single jellyroll fold, which comprises viral capsids and is present in viruses of many genome types. The double jellyroll fold also comprises viral capsids, although it is restricted to dsDNA viruses[21]. RNA viral families often encode nucleocapsids, responsible for packaging of viral RNA, and RNA-dependent RNA polymerases responsible for genome replication. Although the RNA-dependent RNA polymerase is universally conserved in RNA viruses, it is split among multiple protein clusters owing to variation in protein length. By contrast, small dsDNA viruses such as papillomaviruses and polyomaviruses encode a viral replicase with conserved origin binding and helicase domains. Altogether, we find that our structural database successfully reconstitutes conserved viral proteins across diverse viral subtypes.

We next investigated the taxonomic distribution of viral protein clusters. We performed structural alignments of viral protein cluster representatives against 2.3 million cluster representatives from the entire AlphaFold database[3] (Fig. 1h). For each virus protein cluster, we determined the last common ancestor of viruses that encode a cluster member. We found that 29% of protein clusters are present in multiple viral families, the majority of which are present in the AlphaFold database, suggesting that they are evolutionarily ancient (Fig. 1i). In addition, we found that 62% of viral proteins (or 55% of proteins from non-singleton clusters) are restricted to a single viral family and lack analogues in the AlphaFold database (Fig. 1i). This shows that viral evolution generates substantial numbers of novel proteins that are absent from current structure databases.

## Similarities between viral proteins

We investigated the ability of structural alignments to identify relationships that are not apparent from protein sequence alone. We found that many representatives of sequence clusters are structurally similar despite low sequence similarity (Fig. 2a). Adding structural information to protein clustering efforts leads to more taxonomically diverse protein clusters, with significantly more viral families per cluster (Fig. 2b). This is especially important for finding similarity between proteins from divergent viruses, resulting in a substantial increase in protein clusters that encompass proteins encoded by viruses of different genome types (Fig. 2c).

We explored whether structural alignments can link poorly annotated sequence clusters with those that are more annotated (Fig. 2d). We used the sequence-based classifier InterProScan[22] to assign all proteins Pfam[23], Conserved Domain Database[24] (CDD) and TIGR-FAM[25] classifications. Sequence clusters contain almost entirely InterProScan-annotated or entirely unannotated members, resulting in a bimodal distribution of sequence clusters (Fig. 2e). Of the proteins in clusters with more than 1 member, more than 25% of unannotated proteins are located in either an annotated sequence cluster or a protein cluster that contains an annotated sequence cluster (Fig. 2f).

Many protein clusters encompass a mixture of annotated and unannotated sequence clusters (Extended Data Fig. 3a). We find that these connections between sequence clusters are useful to determine putative functions of poorly characterized proteins across the virome. For example, although the single jellyroll fold is the most abundant protein cluster, many members of this cluster are not correctly annotated (Extended Data Fig. 3b). Many other protein clusters include both annotated and unannotated sequence clusters, including clusters encoding enzymes such as nucleotide-phosphate kinases (Extended Data Fig. 3c), NUDIX hydrolases (Extended Data Fig. 3d), DNA ligases (Extended Data Fig. 3f) and nucleases (Extended Data Fig. 3g). One cluster of note includes members that resemble the UL43 family of late herpesvirus proteins (Extended Data Fig. 3e), which will be discussed later.

We next investigated DNA-binding proteins, which have biotechnology applications in diagnostics and genome editing. First, we investigated TATA-binding proteins (TBPs), which bind to TATA-box motifs in eukaryotic promoters[26]. Many DNA viruses target human TBP to promote viral gene expression or modulate host gene expression[27,28]. So far, three families of large dsDNA viruses have been found to encode viral TBPs[29]. We found evidence of these proteins in four additional families of large dsDNA viruses (Fig. 2g), substantially expanding the diversity of virus-encoded TBPs. Next, we investigated the I3L family of single-stranded DNA (ssDNA)-binding proteins encoded by poxviruses (Fig. 2h). I3L potently and specifically binds ssDNA and is thought to be a DNA-binding protein involved in viral DNA replication or repair[30]. There is no experimental structure of I3L, and its link to other protein folds and families remains unknown[30]. We find that I3L contains an oligonucleotide-binding fold (OB-fold), similar to the baculovirus DNA-binding protein DBP and phage T7 single-stranded binding protein (SSB), consistent with the shared ssDNA-binding behaviour of these proteins[31,32]. We confirm the presence of similar OB-fold proteins across four additional dsDNA virus families, showing that Poxvirus I3L represents a widespread family of ssDNA-binding proteins. These eukaryotic dsDNA virus OB-folds contain a distinctive N-terminal beta sheet that is absent in the other baculovirus-encoded OB-fold protein, LEF-3 (Fig. 2i). Together, these results demonstrate that large-scale clustering based on sequence plus predicted structure enables functional inference of poorly characterized viral proteins.

## Similarity to non-viral proteins

Unlike nucleotide or protein sequence, structural features are often conserved over large evolutionary timescales. Thus, we investigated whether alignment between predicted viral and non-viral protein structures can offer insight into the function of poorly annotated proteins encoded by human pathogens. To do this, we used Foldseek to align our virus protein structure database with the initial release of the AlphaFold database, which contains more than 500,000 proteins from 48 organisms across eukaryotes, bacteria and archaea[2] (Fig. 3a). This revealed pervasive structural similarity between viral and non-viral

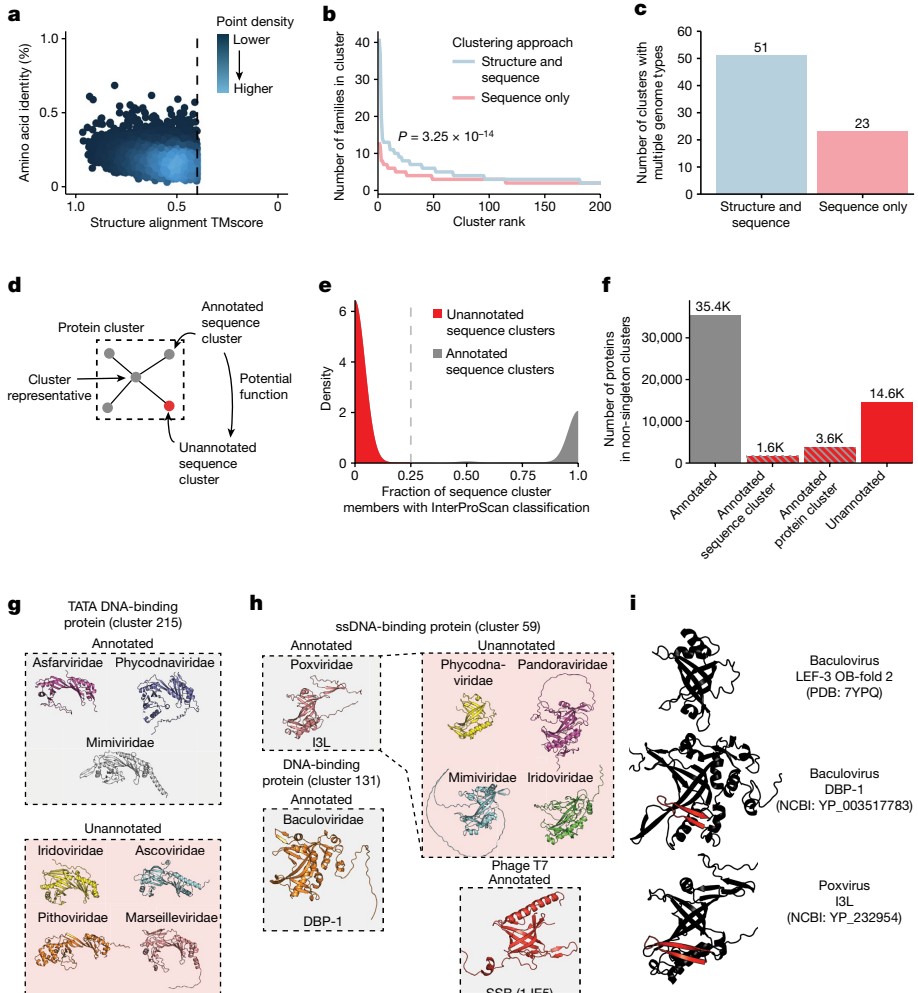

**Fig. 2 | Structural alignments link annotated and unannotated sequence clusters. a**, Structure and sequence similarity between protein cluster representatives. Each dot indicates a single alignment. **b**, Viral family diversity in clusters generated by structure and sequence or sequence alone. The top 200 clusters by number of members were plotted. The *P* value is from a two-sided Wilcoxon rank-sum test. **c**, The number of clusters that contain proteins from viruses with different genome types when using structure and sequence or sequence only. **d**, Structural similarity between InterProScan annotated and unannotated protein clusters has the potential to provide functional information. **e**, The percentage of sequence cluster members with an InterProScan classification is plotted against the density of sequence clusters with each percentage. Sequence clusters with fewer than 25% of members

having InterProScan classifications were considered unannotated sequence clusters. **f**, Counts of proteins annotated by InterProScan or in a protein or sequence cluster with a protein annotated by InterProScan. **g**, Cluster 215 contains TATA DNA-binding proteins. NCBI Protein accessions: YP_009703143, YP_008052367, YP_003969792, YP_009021140, YP_009701471, YP_009000953 and YP_009094710. **h**, Cluster 59 contains a widespread family of ssDNA-binding proteins. NCBI Protein accessions: YP_232954, NP_048769, YP_008437003, YP_003970005, YP_009272775 and YP_003517783. These folds share an oligonucleotide fold with phage T7 single-stranded binding protein. **i**, I3L-like eukaryotic ssDNA-binding proteins contain a distinct N-terminal beta sheet that is absent in other OB-folds such as those present in baculovirus LEF-3.

proteins, with high structural similarity in the face of low amino acid identity (Fig. 3b).

Ultimately, 14,531 predicted viral proteins have an alignment to a member of the AlphaFold database, with the majority of alignments being against proteins encoded by eukaryotes (Fig. 3c). These alignments include proteins that are unannotated but are encoded by human pathogens. To reduce rates of false negatives, we conducted a series of alignments using DALI[19], which is slower than Foldseek but substantially more sensitive. First, we found that a set of proteins encoded by poxviruses are structurally similar to the auto-inhibitory domain of mammalian gasdermins[33,34] (Extended Data Fig. 4a). Similarly, several poxvirus proteins are structurally similar to the human galactosyltransferase COLGALT1, which is thought to enable virus binding to surface glycosaminoglycans during viral entry[35] (Extended Data Fig. 4b). In addition, we observed structural similarity of Poxvirus C4-like proteins with eukaryotic dioxygenases (Extended Data Fig. 4c), consistent with

previous work that identified frequent exaptation of inactivated host enzymes by poxviruses[36]. Vaccinia virus C4 is notable for antagonizing several innate immune pathways. C4 directly binds the pattern recognition receptor DNA-dependent protein kinase (DNA-PK), blocking DNA binding and immune signalling through that pathway[37]. In addition, C4 inhibits NF-κB signalling downstream at or downstream of the IκB kinase (IKK) complex, but the mechanism of this inhibition is unknown[38]. Further studies are required to determine whether its dioxygenase-like fold is involved in its innate immune antagonism.

Next, we found that human herpesviruses UL43-like proteins, including the protein BMRF2 from Epstein–Barr herpesvirus (EBV) and Varicella zoster virus (VZV), share structural similarity with the human equilibrative nucleoside transporter ENT4 (Extended Data Fig. 4d). We conducted structural alignments using DALI of EBV BMRF2 against proteins classified in the Transporter Classification Database[39] (TCDB) (Fig. 3d). This revealed that BMRF2 has strong structural

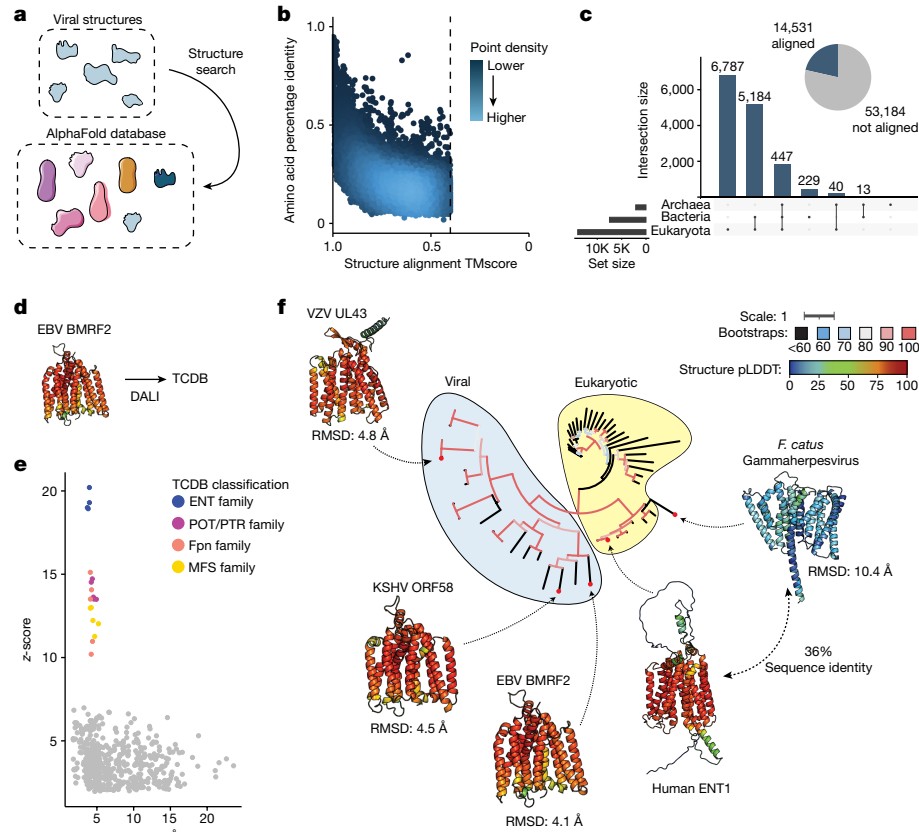

**Fig. 3 | Structural similarity across kingdoms of life reveals potential protein function. a**, Illustration of the approach. The database of viral protein predicted structures was aligned against the AlphaFold database of proteins from 48 organisms, including members of the bacterial, eukaryote and archaeal superkingdoms. **b**, The amino acid percentage identity and Foldseek TMscore; each point indicates a single alignment. For viral proteins with more than five alignments, the top five alignments by TMscore are plotted. **c**, Right, pie chart indicating the number of viral proteins that do or do not have an alignment against the AlphaFold database. Left, UpSet plot indicating, for those viral proteins with alignments against the AlphaFold database, the number that align against members of each superkingdom. **d**, EBV BMRF2 (YP_001129455), which has a nucleoside transporter-like fold, was used as a query for a DALI

search against the TCDB. **e**, Alignments between EBV BMRF2 and structures classified in the TCDB. Each dot indicates a single DALI alignment. Proteins with at least one alignment with $z \geq 10$ are coloured. RMSD, root mean squared deviation as determined by DALI. **f**, A phylogenetic tree of eukaryotic and herpesvirus nucleoside transporters. The listed RMSD values were determined by DALI alignment between human ETN1 and each viral nucleoside transporter. The tree scale is substitutions per residue. Structures are coloured by pLDDT (red, higher; blue, lower). The tree is coloured according to bootstrap values. Accessions: *F. Catus* gammaherpesvirus, YP_009173937; VZV UL43, NP_040138; EBV BMRF2, YP_001129455; KSHV ORF58, YP_001129415; human ENT1, XP_011512643.

similarity to human equilibrative nucleoside transporter (ENT)-family transporters, with weaker similarity to related transporter families (Fig. 3e). We generated a phylogenetic tree of herpesvirus UL43-like proteins and related eukaryotic proteins, revealing that these proteins are widely distributed across herpesviruses (Fig. 3f). Notably, we identify a variant encoded by *Felis catus* gammaherpesvirus that maintains 36% sequence similarity to human ENT1, supporting the structural connection between these herpesviral proteins and ENT proteins. EBV substantially remodels host cell metabolism during viral infection[40], and this finding suggests a potential metabolic role in addition to BMRF2 involvement in viral attachment[41]. In addition, transport of antiviral nucleoside analogues such as valacyclovir are mediated by nucleoside transporters[42], raising questions about the interplay between this protein and valacyclovir during VZV infection. These proteins belong to a cluster of proteins similar to the UL43 family of late herpesvirus proteins, some of which are unannotated (Extended Data Fig. 3e). Further experimental characterization is required to confirm the substrate(s) that are transported by these putative transporters. Together, these findings illustrate the ubiquity of structural similarity between viral and non-viral proteins and show that this similarity can be used to predict potential functions of poorly characterized viral proteins.

Although we found that some protein clusters contain members encoded by viruses of different genome types, the evolutionary origin of such conservation is unclear. Many of these protein clusters are predominantly encoded by viruses of a single genome type but are expressed in a small minority of viruses of a different genome type (Extended Data Fig. 5a). This observation is consistent with virus–virus or host–virus horizontal gene transfer. To explore this possibility, we conducted searches of sequence cluster representatives against viral and non-viral protein databases and constructed phylogenetic trees of the top hits. We found that nucleoside-phosphate kinases in cluster 28 show a polyphyletic distribution with homologues in different viruses showing amino acid similarity to distinct sets of non-viral proteins (Extended Data Fig. 5b). There is a similar pattern with HrpA/B-like helicases in cluster 55, with helicases in different viral families exhibiting amino acid similarity to distinct sets of non-viral organisms (Extended Data Fig. 5c). These patterns are consistent with horizontal gene transfer from non-viral hosts. By contrast, other taxonomically distributed protein clusters such as cluster 56 (encoding parvovirus Rep proteins with homologues in some human herpesviruses) and cluster 735 (encoding a haemagglutinin lineage that is present in baculoviruses and some orthomyxoviruses) display a monophyletic taxonomic distribution consistent with horizontal gene transfer

between viruses (Extended Data Fig. 5d,e). These data suggest that many protein clusters that contain proteins from viruses of different genome types arise from horizontal gene transfer from both viral and non-viral sources.

## Identification of shared domains

We constructed protein clusters with a strict 70% coverage requirement, leaving open the possibility that individual domains can be identified through structure comparison[3]. We reasoned that protein domains that are present within multiple protein clusters may have particular biological importance. We used DALI to conduct all-by-all alignments of the representatives of all protein clusters having more than one member. This revealed substantial protein similarity with many alignments having $z$-scores greater than 8 (Extended Data Fig. 6a). Protein clusters ultimately fall into a network of shared domains (Extended Data Fig. 6b). Here we find that distinct domains are often shared across protein clusters in context with various combinations of other domains, which can be seen with domains involved in interaction with the cytoskeleton (Extended Data Fig. 6c) and in metabolism (Extended Data Fig. 6d,e) in eukaryotic viruses and phage.

## Sensitivity of structural searches

We compared the sensitivity of our approach, which uses both sequence and structure, to methods that use only sequence information. First, we investigated the ability of sequence methods to reconstitute our viral protein clusters. For all protein clusters with at least two sequence clusters, we conducted all-by-all alignment with three different sequence methods. We then used connected-component clustering to identify clusters based on these methods (Extended Data Fig. 7a). We found that sequence methods fail to group all proteins into a single cluster (Extended Data Fig. 7b,c); jackhmmer, for example, identifies an average of more than two clusters for each single protein cluster generated by our sequence and structure method.

We next investigated the ability of sequence methods to identify similarities between viral and non-viral proteins. We first conducted sequence searches analogous to the DALI searches between non-viral and viral structures conducted in Extended Data Fig. 4, and quantified the fraction of the DALI alignments that are reconstituted by each sequence method. These sequence methods were unable to identify the vast majority of alignments (Extended Data Fig. 7d). Next, we conducted a broader quantitative comparison between DALI and hhPred, a highly sensitive sequence-based method[43]. We identified 4,409 non-singleton sequence clusters that contained fewer than one-quarter of members with an Interproscan alignment. Of these clusters, 1,326 had a well-folded cluster representative with an average pLDDT of at least 70. We used DALI to align each of these structures against the PDB25 database, a sequence-clustered subset of the PDB provided by the DALI authors. In addition, we established a local version of hhPred, which is a two-step approach using HHblits and HHsearch, and used this pipeline to search the amino acid sequences of each of these proteins against the PDB. This analysis revealed that DALI was able to identify confident alignments for 661 out of 1,326 proteins, compared to just 295 by HHsearch (Extended Data Fig. 7e,f). Together, these data show that structural methods on proteins with high-quality structure predictions often outperform sequence methods at identifying similarities between viral proteins and other viral or non-viral proteins.

## Discovery of cGAMP PDEs

Many aspects of eukaryotic and prokaryotic immunity have a shared origin[44]. One set of related pathways are the mammalian cyclic GMP-AMP synthase (cGAS)–STING and oligoadenylate synthase (OAS) pathways and prokaryotic cyclic-oligonucleotide-based anti-phage signalling

systems (CBASS). In both cases, a protein sensor detects a viral cue and generates a nucleotide second messenger, which activates a downstream antiviral effector (Fig. 4a). In the case of the cGAS pathway, cGAS recognizes cytoplasmic dsDNA and generates 2′,3′-cGAMP. Many cGAS/DncV-like nucleotidyltransferases (CD-NTases) in prokaryotic CBASS systems make a similar second messenger, 3′,3′-cGAMP, in response to viral cues[45]. By contrast, OAS recognizes double-stranded RNA (dsRNA) and generates linear 2′,5′-oligoadenylates[46]. In prokaryotes, phage T4 encodes the LigT-like PDE anti-CBASS protein 1 (Acb1), which degrades 3′,3′-cGAMP and a variety of other cyclic nucleotide substrates including 2′,3′-cGAMP[47].

In eukaryotes, several RNA viruses encode PDEs that degrade 2′,5′-oligoadenylates[48,49]. Notably, we find that these PDEs have a LigT-like fold similar to Acb1. Given the conserved use of LigT-like PDEs in viral anti-immunity, we investigated their distribution and phylogeny. Structural searches revealed that many different branches of LigT-like PDEs are present in eukaryotic viruses (Fig. 4b). Notably, there are multiple independent branches of LigT-like PDEs in RNA viruses. Linage A betacoronaviruses and toroviruses share a clade of PDEs that is similar to the PDEs present in rotaviruses. Surprisingly, lineage C betacoronaviruses contain a distinct branch of PDEs[49] (Fig. 4b). This suggests that there were two independent PDE acquisition events within the betacoronavirus genus, showing the strong selective pressure for betacoronaviruses to evade the OAS pathway. We find that some large DNA viruses also contain LigT-like PDEs. Despite the extreme amino acid variability across the LigT-like PDE tree, there is near-universal conservation of the two catalytic histidines across viral LigT-like PDEs.

The presence of LigT-like PDEs in large DNA viruses raises the question of whether they have an anti-immune function. Whereas the RNA-sensing OAS pathway is commonly targeted by LigT-like PDEs of RNA viruses, there is likely to be less pressure for large DNA viruses to target OAS. Thus, we tested whether LigT-like PDEs encoded by large DNA viruses have activity against 2′,3′-cGAMP. First, we cloned and tested the expression of a panel of LigT-like PDEs, and found a subset that can be expressed well in mammalian cells (Extended Data Fig. 8a). Next, we generated a synthetic STING circuit in 293T cells[50] (Fig. 4c). In this system, STING can be activated by treatment with cGAMP or the non-nucleotide STING agonist diABZI[51], which leads to expression of firefly luciferase in a STING-dependent manner. We expect that a viral LigT that targets cGAMP should be able to inhibit cGAMP- but not diABZI-mediated STING activity. Testing well-expressing LigTs revealed that LigT-like PDEs encoded by avian poxviruses have very potent activity against 2′,3′-cGAMP-mediated STING signalling but have limited activity against diABZI-mediated STING signalling (Extended Data Fig. 8b). Furthermore, mutation of the catalytic histidines of the LigT-like PDEs substantially reduces activity (Extended Data Fig. 8b). Next, we tested the activity of the pigeonpox LigT against a panel of cGAMP isomers, including 2′,3′-, 3′,3′- and 3′,2′-cGAMP. This revealed that, similar to T4 Acb1, pigeonpox LigT has widespread activity against diverse cGAMP variants (Fig. 4d).

To confirm that pigeonpox LigT degrades cGAMP variants, we purified wild-type and mutant (H72A/H167R) pigeonpox PDE and visualized cleavage of 2′,3′- and 3′,3′-cGAMP by TLC. Similarly to phage T4 Acb1 and unlike the 2′,5′-oligoadenylate-targeting LigT-like PDE NS2a from murine hepatitis virus (MHV), pigeonpox PDE cleaves 2′,3′- and 3′,3′-cGAMP (Fig. 4e). Furthermore, 2′,3′- and 3′,3′-cGAMP degradation by pigeonpox PDE and phage T4 Acb1 result in products that co-migrate on TLC, indicating a conserved mechanism of cGAMP hydrolysis by the two enzymes (Extended Data Fig. 8c). Avian poxviruses are notable for their lack of poxin[52,53], the other 2′,3′-cGAMP PDE encoded by poxviruses, showing the strong selective pressure for poxviruses to evade cGAS–STING immunity. These results leave open the possibility that other lineages of LigT-like PDEs in large dsDNA viruses may have cGAMP activity. In sum, we have leveraged structure similarity to discover a novel mechanism of 2′,3′-cGAMP degradation by eukaryotic

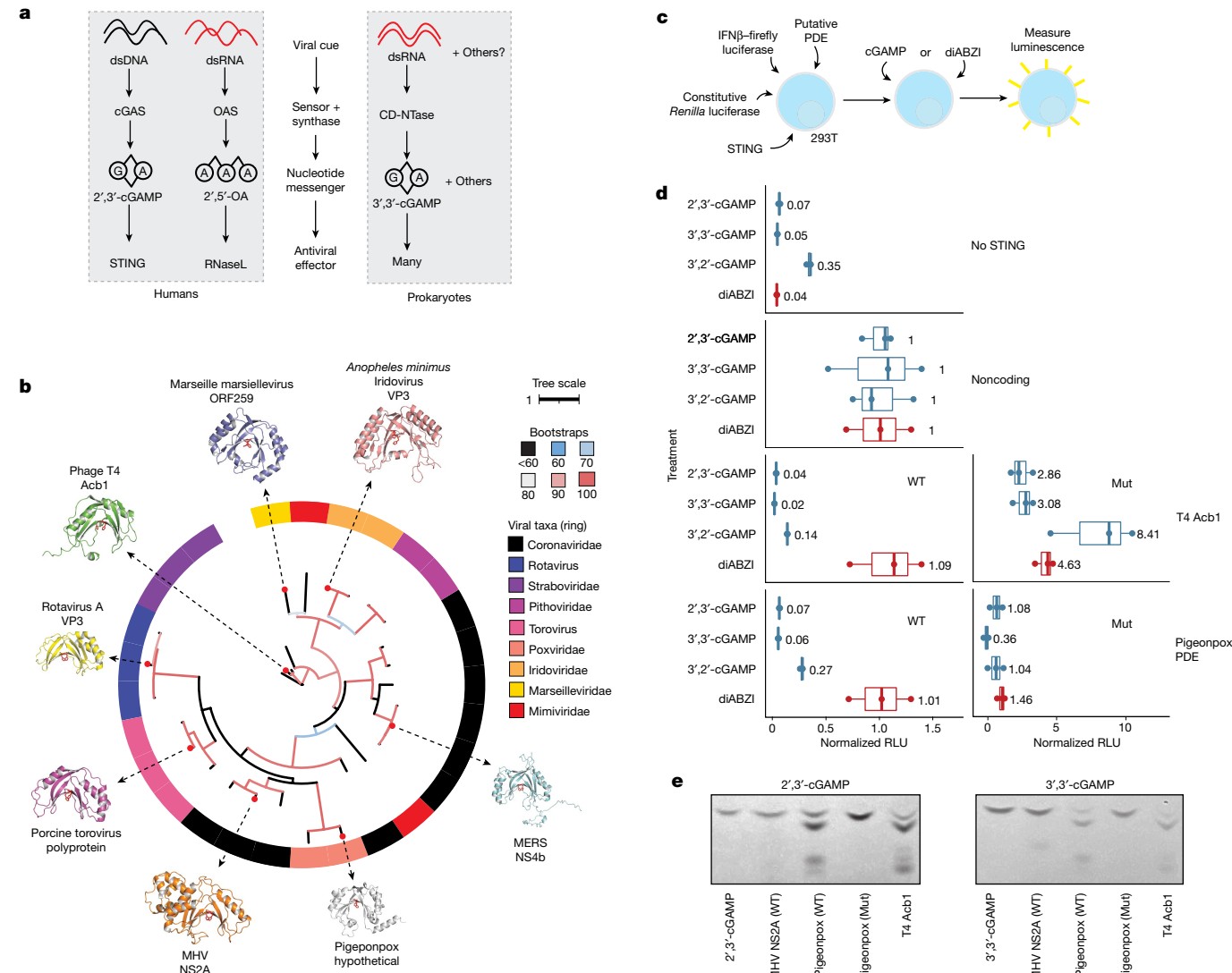

**Fig. 4 | LigT-like PDEs are frequently used to subvert host immunity. a**, Some innate immune pathways in eukaryotes and prokaryotes rely on a viral synthase sensor that detects virus-associated molecular patterns such as dsDNA or dsRNA and generates a nucleotide second messenger that stimulates an antiviral effector. **b**, A phylogenetic tree showing the polyphyletic lineages of LigT-like PDEs. Shaded boxes indicate viral taxa. The red residues in each protein structure are the conserved catalytic histidines. Units are substitutions per residue. The tree is coloured according to bootstrap values. NCBI Protein accessions: YP_008798230, YP_002302228, YP_009021100, YP_003406995, NP_049750, YP_009047207, YP_009046269 and YP_009824980. **c**, HEK 293T cells were transfected with constructs encoding STING, firefly luciferase driven by an *IFNB* promoter, a constitutively expressed *Renilla* luciferase, and a transgene. After 5 h, cells were treated with 10 μg ml⁻¹ cGAMP or 0.1 μM diABZI.

Around 24 h after the first transfection, luminescence of the firefly and *Renilla* luciferases was measured. **d**, Pigeonpox PDE prevents STING activation by cGAMP isomers. On the *x* axis, luminescence in relative luminescence units (RLU) is normalized to the RLU from cells transfected with noncoding vector and treated with the same STING agonist. RLUs were initially normalized as firefly RLU/*Renilla* RLU. Mut indicates mutations of the catalytic histidines. In box plots, the centre line is the median, box edges delineate 25th and 75th percentiles, and whiskers extend to the highest or lowest point up to 1.5 times the inter-quartile range. Data are from one biological replicate and three wells per condition. **e**, 2′,3′-cGAMP or 3′,3′-cGAMP was incubated with indicated wild-type or catalytic histidine mutant PDE proteins. Degradation of each cGAMP isomer was visualized by TLC. Uncropped TLC images are presented in Supplementary Fig. 1.

viruses and find that cGAMP targeting by LigT-like PDEs is a pan-viral mechanism of anti-immunity.

## Discussion

Viruses have yielded fundamental insights into basic molecular biology. Here we cluster viral proteins and use structural alignments to gain functional insights that could not be obtained with prior approaches. We have raised testable hypotheses about the function of proteins present in human pathogens, and provide a resource for studying viral protein structures at scale. Expanding databases of structures and predicted structures will continue to enable functional inference.

This is important not only from a fundamental biology perspective, but also in light of the continual emergence of novel viruses with pandemic potential. Structural similarity both with other viral proteins and with host proteins can offer functional insights and provide insight into the origin and evolution of viral proteins. A caveat of our study is the use of a stringent 70% coverage threshold during clustering. This means that some proteins with similar function but differences in domain configuration will be split into separate protein clusters, underestimating their taxonomic diversity. However, we find that structural alignment-based domain identification can identify shared structural repeats and enable comparison across protein structures. Both structure prediction and alignment have fundamental limitations.

These structures are predictions, whose quality can vary and can be influenced by the depth of the MSA used for prediction. Structural alignments in turn can be affected by the arbitrary positioning of protein domains.

Protein structure is especially informative in cases of evolutionary distance. One impactful area is the concept of conserved viral anti-immunity. Emerging evidence on the common origin of some bacterial and eukaryotic immune systems raises the potential for conserved anti-immunity systems in eukaryotic viruses and phage. This is illustrated by LigT-like PDEs, which have been adapted multiple times by phage and eukaryotic viruses to evade innate immunity. This also illustrates the flexibility of core protein folds, as this conserved fold can be adapted to cleave distinct immune second messengers depending on the pathway being targeted. Together, our study lays the foundation for characterization of viral protein evolution and function across the virome.

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

## Methods

### Preparation of protein sequences

Protein sequences for eukaryotic viruses present in RefSeq[54] were collected through the NCBI Viruses portal (https://www.ncbi.nlm.nih.gov/labs/virus) in July 2022. GenPept files were downloaded for viruses that were annotated by NCBI to have an eukaryotic host. Because not all viruses have a host labelled by NCBI, GenPept files of human-infecting viruses annotated by ViralZone (https://viralzone.expasy.org/678) were also downloaded. Finally, proteins from all coronaviruses present in RefSeq, regardless of NCBI-labelled host, were downloaded.

Each GenPept file was processed such that polyproteins with defined 'mature peptide' fields produced separate protein sequences for each mature peptide. GenPept files without a mature peptide field were output as full amino acid sequences. These processing steps are present in the vpSAT github directory (https://github.com/jnoms/vpSAT) in the process_gbks.py file. Proteins larger than 1,500 residues, or in some cases 1,000 residues, were excluded. Only 1,706 proteins were excluded for this reason.

### Structure prediction

MSAs were generated with MMseqs2 release version b0b8e85f3b-8437c10a666e3ea35c78c0ad0d7ec2. To increase MSA generation speed, the RefSeq virus protein database (downloaded on 6 June 2022) was used as the target database for MSA generation. Structures were predicted with ColabFold[15] (downloaded 22 June 2022). The majority of samples used three recycles, three models, stop_at_score=70, and stop_at_score_below=40. MMseqs2 and Colabfold_batch were run with a Nextflow[55] pipeline, and all parameters used can be found at https://github.com/jnoms/vpSAT. Information on all viruses and structures included in this manuscript is present in Supplementary Table 1.

### Protein cluster generation

All proteins were initially clustered with MMseqs2, with a requirement of at least 20% sequence identity and 70% query and target coverage. MMseqs2 cluster mode 0 was used, meaning that many but not all pairs of aligned proteins are placed into the same sequence cluster. Predicted structures for each sequence cluster representative were subjected to an all-by-all alignment using Foldseek[17], requiring the alignment to consist of at least 70% query and target coverage and an alignment $E$-value less than 0.001. The resultant structural alignment file was then filtered using SAT aln_filter to keep alignments with a TMscore of at least 0.4. Clusters were generated from this alignment file using SAT aln_cluster in a similar manner as Foldseek cluster mode 1, wherein all query-target pairs are assigned to the same cluster. Cluster information from sequence and structure clustering were merged using SAT aln_expand_clusters. Taxonomic counts information was generated using SAT aln_taxa_counts, producing a 'tidy' table for each cluster_ID with the number of members of each taxon at multiple taxonomy levels. Taxonomy information was also added directly to the merged cluster file using SAT aln_add_taxonomy.

### Cluster purity analysis

To determine the structural consistency of the clusters, all clusters with at least 100 members were selected for analysis. DALI was used to align the cluster representative with each cluster member. Clusters whose members were on average smaller than 150 residues were excluded. This led to the analysis of 49 clusters. Cluster members that failed to align to their representative were assigned a $z$ value of 0. For each cluster, the average $z$-score between the representative and each member was determined and plotted. All scripts used to run DALI can be found in vpSAT's dali_format_inputs.sh and dali.sh files. Dalilite version 5 was used. DALI output files were parsed into a tabular format using SAT's aln_parse_dali.

### Phylogenetics

Phylogenetic reconstructions were conducted using all sequence cluster representatives, or in the cases of clusters 56 and 735, all members within each cluster. For the nucleoside transporter tree, all herpesvirus sequence representatives of cluster 119, as well as a *F. catus* gammaherpesvirus 1 protein (YP_009173937) from a singleton cluster, were used as queries. Iterative sequence similarity searches against the NCBI non-redundant database were performed using standalone PSI-BLAST v2.15.0, using the following parameters[56]: -num_iterations 10, -max_hsps 1, -subject_besthit, -gapopen 9, -inclusion_ethresh 1e-15, -evalue 1e-10, and -qcov_hsp_perc 70. For the LigT-like PDE tree, this search was restricted to only viral targets. Each of these protein sets were then clustered by utilizing mmseqs2 v15.6f452 with high sensitivity (command line option: -s 7.5) to compress the amount of highly similar sequences into cluster representatives. Subsequently, these sequence sets were aligned using Clustal Omega v1.2.4 with default settings[57]. Comprehensive taxonomic information for each aligned sequence was integrated into the unique sequence identifiers by utilizing the biopython v1.81 package[58]. Phylogenetic trees were reconstructed using IQTREE v2.3.3[59] with -m TEST -B 1000 options for model testing and bootstrapping. The best model was selected for each tree based on Bayesian Information Criterion (BIC), and were as follows: Nucleoside transporters, VT + F + G4; LigTs, VT + F + G4; cluster 28, VT + G4; cluster 55, VT + I + G4; cluster 56, VT + G4; cluster 735, VT + I + G4. Trees were visualized with the Interactive Tree of Life (iTOL)[60]. Code used for this analysis can be found at https://github.com/Doudna-lab/nomburg_j-LigT_phylogeny.

### Structural alignments against the AlphaFold databases

In Fig. 1i, Foldseek was used to align a protein representative from every viral protein cluster against 2.3 million protein cluster representatives from the AlphaFold database[3]. For Fig. 3, all 67,715 viral protein structures were searched against the pre-made Foldseek databases of the original release of the AlphaFold database (downloadable via the Foldseek command 'foldseek databases Alphafold/Proteome afdb tmp'), consisting of proteins from 48 organisms and including members of the bacterial, eukaryote, and archaeal superkingdoms. For this search, the full AlphaFold database of over 200 M structures was not used because it contains many viral proteins misannotated as non-viral proteins (these misannotations reflect errors in Uniprot metadata). Alignments were filtered to keep only those with a minimum TMscore of 0.4 and an $E$-value of less than 0.001.

### DALI alignments of specific non-viral proteins against the viral protein database

Following Foldseek alignments against the AlphaFold database, specific hits of interest (for example, ENT4) were selected. These structures were downloaded and imported to the DALI database format using vpSAT's dali_format_inputs.sh. They were then aligned against the full viral protein structure database using vpSAT's dali.sh, which lists all parameters. Dalilite version 5 was used. DALI output files were parsed into a tabular format using SAT's aln_parse_dali.

### Identification of annotated protein sequence clusters

Each protein in the database was searched against the Pfam[23], CDD[24], and TIGRFAM[25] databases using InterProScan[22]. A sequence cluster was considered annotated if more than 25% of members had any InterProScan alignment, and was considered unannotated if otherwise. Note that some proteins without an InterProScan alignment have existing annotations through other methods, including manual curation. Values of RMSD in Fig. 3 were calculated using DALI.

## DALI alignments to identify shared domains

This analysis used the structure representatives from clusters with at least 2 members, resulting in 5,700 cluster representatives. Structures from these representatives were imported to the DALI database format using vpSAT's dali_format_inputs.sh. To compare eukaryotic virus protein cluster representatives, an all-by-all alignment was conducted using vpSAT's dali.sh, which lists all parameters.

Dalilite version 5 was used. DALI output files were parsed into a tabular format using SAT's aln_parse_dali. All DALI alignments were filtered for an alignment length of at least 120, and for a $z$-score greater than or equal to (alignment length/10) − 4.

## MSA generation using the full ColabFold MMseqs2 database

We selected the protein cluster representatives from the top 100 protein clusters by size, as well as 100 randomly selected singleton clusters, for analysis. ColabFold was used with FASTA inputs, such that MSAs were generated using the MMseqs2 ColabFold server (which maps each sequence against UniRef, BFD and Mgnify), and this MSA was used for structure prediction.

## Benchmarking sequence and structure methods

For all protein clusters with at least two sequence clusters, we conducted all-by-all alignments between members using MMseqs2 (version b0b8e85f3b8437c10a666e3ea35c78c0ad0d7ec2), DIAMOND blastp[61] (version 0.9.14), or jackhmmer[62] (version 3.1b2). These alignments and subsequent clustering occur separately for each protein cluster. From these alignments, we conducted connected-component clustering using sat.py aln_cluster. Here, all proteins that align will be assigned to the same resultant cluster. Thus, each original protein cluster (determined through our approach, combining sequence alignment with MMseqs2 and structure alignment with Foldseek) now has a set of clusters identified through each of the sequence-only methods. We then measured, for each original protein cluster, how many clusters created by each of the sequence-only methods and how many proteins fall into the largest cluster generated by these sequence methods.

For benchmarking virus–non-virus alignments, we conducted sequence alignments (again using MMseqs2, DIAMOND blastp, and jackhmmer) analogous to the DALI structural alignments present in Extended Data Fig. 4, using the same query against all viral proteins included in the dataset. We then determined the fraction of DALI-identified targets were identified for each non-viral query and through each sequence method.

For the comparison between hhPred[43] and DALI, we identified 4,409 sequence clusters that contained more than 1 member and for which fewer than one-quarter of members had an InterProScan alignment. We then identified sequence cluster representatives that were well folded, with an average pLDDT of at least 70. This resulted in a final set of 1,326 proteins. We used DALI to align each of these proteins against the PDB25 database provided by the DALI authors. Alignments were considered high-confidence if they contained a $z$-score of at least 7. DALI alignments were conducted with vpSAT's dali.sh. For hhPred searches, we established a local pipeline using HHsuite's (v3.3.0) HHblits and HHsearch modules. For each query protein, we first used HHblits to align them against the Uniref30 HMM database provided by the HHsuite authors, using the flags -n 2 and -cov 20. We then used HHsearch to align each resultant MSA against the HHsuite-provided PDB database with the flag -cov 20. Alignments were considered high-confidence if they had an $E$-value of less than or equal to 0.001.

## Searching the TCDB

We used a map of PDB accession to TCDB classification (https://www.tcdb.org/cgi-bin/projectv/public/pdb.py) to download all experimental structures associated with TCDB classifications. For subsequent processing, we used a maximum of five structures per TCDB classification. One structure was excluded (PDB: 1HXI) as it is highly truncated. Nine additional structures failed to import to DALI database files, typically due to small protein size. For PDB entries that contained multiple chains, we selected the first chain for alignment. Due to the absence of experimental structures, the AlphaFold models for ENT3 (AF-Q9BZD2-F1-model_v4) and ENT4 (AF-Q7RTT9-F1-model_v4) were added to the dataset. For the 46 protein structures with multiple classifications, one classification was chosen at random. This ultimately resulted in a dataset of 1,812 structures from 485 classifications, with an average of 3.7 structures per classification. Structures were imported to the DALI database format using vpSAT's dali_format_inputs.sh. The predicted structure of EBV BMRF2 (YP_001129455) was aligned against this structure database using dali.sh.

## PDE cloning and activity assays

Two tandem STREP2 tags, following a GGS linker, were appended to the end of each putative LigT-like PDE. Sequences were codon-optimized for humans, and gBlocks encoding each product were ordered from IDT and cloned into a custom lentiviral expression vector. PDE mutants have dual H>A mutations of the catalytic histidines (or, in the case of MHV NS2a and pigeonpox PDE, one H>A and one H>R mutation).

The 293T cells were seeded into 96-well plates at 20,000 cells per well. The 293T cells were kindly provided by the Ott laboratory, and were originally from ATCC. The 293T cells were screened for *Mycoplasma* within the last year, and were not otherwise authenticated. The day after plating, each well was transfected with 15 ng STING (pMSCV-hygro-STING R232, Addgene 102608), 20 ng firefly luciferase driven by an *IFNB* promoter (IFN-Beta_pGL3, Addgene 102597), 5 ng *Renilla* luciferase (pRL-TK, Promega E2241), and 20 ng of each putative PDE using the Mirus TransITX2 transfection reagent. After at least 4 h, cells were treated with 0.1 µM diABZI (Invivogen) or transfected with 10 µg ml$^{-1}$ 2′,3′-cGAMP (Invivogen) using TransITX2. The next day, firefly and *Renilla* luciferase were measured using the Promega Dual-Glo luciferase assay system. Three wells were transfected per condition, and experiments are representative of at least two independent experiments. The 'no STING' conditions were transfected with both reporters and a noncoding transgene, but no STING plasmid.

## PDE western blots

The 293T cells were plated in 6-well dishes at $5 \times 10^5$ cells in 2 ml per well. The next day, each well was transfected with 200 ng of the indicated transgene using Mirus TransITX2. The following day, cells were lysed using RIPA buffer (ThermoFisher) supplemented with protease/phosphatase inhibitor (ThermoFisher), and lysate protein concentrations were determined using the Pierce BCA assay kit. All samples were then normalized to the same protein concentration. Bio-Rad Criterion 4%–20% acrylamide gels were loaded with 30 µg of protein per well, followed by transfer to a 0.2-µm nitrocellulose membrane. For visualization of the Strep-tagged PDEs, the Streptactin HRP (IBA 2-1502-001IAB) antibody was used (1:100,000 dilution, 1 h at room temperature). For visualization of GAPDH, we used Santa Cruz Biotech Mouse anti GAPDH (sc-365062) primary (1:1,000 dilution, incubation at 4 °C overnight) and ECL Anti-mouse IgG (Amersham NXA931) secondary (1:5,000 dilution, 1 h at room temperature).

## Recombinant protein expression and purification

Expression plasmids for pigeon poxvirus PDE (wild-type and H72 A/H167R), MHV nonstructural protein 2A (NS2A), and T4 anti-CBASS protein 1 (Acb1) were cloned into custom pET-based vectors by Gibson assembly to yield N-terminal His$_{10}$-MBP-TEV constructs. Proteins were expressed from 4 l *Escherichia coli* Rosetta 2 (DE3) pLysS by growing to an of OD$_{600}$ of 0.4–0.6 in 2× yeast extract tryptone medium at 37 °C and induced with 0.5 mM isopropyl β-D-1-thiogalactopyranoside. After induction, cells expressing each protein were grown overnight

at 16 °C to an $OD_{600}$ of 1.2–1.4. Cells were collected by centrifugation for 20 min at 4,000 rpm at 4 °C and resuspended in 20 mM Tris-HCl, pH 8.0, 10 mM imidazole, 2 mM $MgCl_2$, 500 mM KCl, 10% glycerol, 0.5 mM TCEP and Roche protease inhibitor. Cells were lysed by sonication and cell lysate was clarified by centrifugation at 17,000$g$, 4 °C for 0.5 h. The supernatant was bound to 5 ml Nickel-NTA affinity resin for 1 h at 4 °C. Supernatant was discarded and resin was washed 5 × 30 ml wash buffer (20 mM Tris-HCl, pH 8.0, 500 mM KCl, 30 mM imidazole, 10% glycerol and 0.5 mM Tris(2-carboxyethhyl) phosphate). Protein was eluted in 10 ml elution buffer (20 mM Tris-HCl, pH 8.0, 500 mM KCl, 300 mM imidazole, 10% glycerol, and 0.5 mM Tris(2-carboxyethyl) phosphate). Each protein was concentrated to 10 mg $ml^{-1}$ during buffer exchange to storage buffer (20 mM Tris-HCl, pH 8.0, 500 mM KCl, 30 mM imidazole, 10% glycerol and 0.5 mM Tris(2-chloroethyl) phosphate) using a 10 kDa MWCO centrifugal filter (Amicon). A total of 5–15 mg target protein fused to N-terminal $His_{10}$–MBP–TEV was stored at −80 °C.

### In vitro characterization of PDEs

Recombinant enzymes were assessed for PDE activity by in vitro cGAMP degradation reactions and downstream analysis by TLC. Reactions were initiated by the addition of recombinant enzyme (40 µM) in reaction buffer (50 mM Tris, pH 8.0, 10 mM $MgCl_2$, 100 mM NaCl) to 1.25 mM 2′,3′-cGAMP or 3′,3′-cGAMP (Biolog). The reaction mixture was incubated at 37 °C for 18 h and stopped by vortexing for 20 s.

Silica gel TLC plates (5 cm × 10 cm) with fluorescent indicator 254 nm were spotted with 2 µl in vitro enzymatic reaction. Separation was performed in an eluent of n-propanol/ammonium hydroxide/water (11:7:2 v/v/v). The plate was allowed to dry fully and visualized with a short-wave ultraviolet light source at 254 nm.

### Data analysis and plotting

All analysis, plotting, and statistical tests used R version 4.0.3. The genome type and average genome size were determined from information downloaded from the NCBI Virus portal (https://www.ncbi.nlm. nih.gov/labs/virus/vssi/#/).

### Reporting summary

Further information on research design is available in the Nature Portfolio Reporting Summary linked to this article.

### Data availability

There are several options for viewing, downloading, and searching the structural models generated here. Searching: (1) We have established a Google Colab notebook that enables any user to quickly and easily search one or more protein structures against our viral structure database using Foldseek (https://colab.research.google.com/ github/jnoms/vpSAT/blob/main/bin/colab/QueryStructures.ipynb). This notebook runs rapidly and displays alignment results and information on the protein clusters to which alignment targets belong. (2) For users who want to conduct high-throughput searches, we have released a pre-made Foldseek database to facilitate use (https://doi. org/10.5281/zenodo.10685504 (ref. 63)). Viewing and downloading: (1) We have established a Google Colab notebook that allows users to explore our data. Users can input a virus taxonomy ID or family name and browse available proteins. Users can then automatically view and download individual structures (https://colab.research.google.com/ github/jnoms/vpSAT/blob/main/bin/colab/ExploreStructures.ipynb). (2) We have uploaded our structures to ModelArchive (https://www. modelarchive.org/doi/10.5452/ma-jd-viral); ModelArchive hosts predicted structures in a uniform way with extensive metadata. Furthermore, ModelArchive is part of the EBI 3D-Beacons framework (https:// www.ebi.ac.uk/pdbe/pdbe-kb/3dbeacons/), which enables uniform downloads and processing of our protein structures through a shared API encompassing the PDB, AlphaFold database, and other databases.

(3) Structures can be accessed through each viral family phage in Viralzone (https://viralzone.expasy.org/10977). (4) Finally, all structures are available on Zenodo (https://doi.org/10.5281/zenodo.10291581 (ref. 64)). Source data are provided with this paper.

### Code availability

Code used for upstream processing is present in the vpSAT Github repository (https://github.com/jnoms/vpSAT/tree/main). This includes scripts required for most computational steps. A workflow is available that shows all main processing steps (https://github.com/ jnoms/vpSAT/blob/main/manuscript_code/latest/analysis_workflow. ipynb). The stable vpSAT version used for this work is available through Zenodo (https://doi.org/10.5281/zenodo.10373132)[65]. The SAT python package can be downloaded as instructed on the SAT Github repository (https://github.com/jnoms/SAT/tree/main). The stable SAT version used for this work is available through Zenodo (https://doi.org/10.5281/ zenodo.10373132)[65]. Code used for phylogenetics analysis can be found here: https://github.com/Doudna-lab/nomburg_j-LigT_phylogeny. All plotting and analysis scripts are available as Quarto documents: https://github.com/jnoms/vpSAT/blob/main/manuscript_code/ latest/. All code can also be found on Zenodo, along with all intermediate data necessary to reproduce the figures (https://doi.org/10.5281/ zenodo.10373132)[65].

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

**Acknowledgements** J.A.D. is an investigator of the Howard Hughes Medical Institute, and research in the Doudna laboratory is supported by the Howard Hughes Medical Institute (HHMI), NIH/NIAID (U54AI170792, U19AI135990, UH3AI150552 and U01AI142817), NIH/NINDS (U19NS132303), NIH/NHLBI (R21HL173710), NSF (2334028), DOE (DE-AC02-05CH11231, 2553571 and B656358), Lawrence Livermore National Laboratory, Apple Tree Partners (24180), UCB-Hampton University Summer Program, Mr. Li Ka Shing, Koret-Berkeley-TAU, Emerson Collective and the Innovative Genomics Institute (IGI). Portions of this research were conducted on the Wynton Cluster at UCSF, supported by UCSF IT. The authors thank members of the Doudna laboratory for their support and helpful feedback, and the James B. Pendleton Charitable Trust. E.E.D. was supported by NIGMS of the NIH under award number F32GM153031.

**Author contributions** J.N. and J.A.D. designed and conceived this project. Generation of the predicted structures, all computational work and all figure generation was performed by J.N., except when noted otherwise. E.E.D. designed and performed all biochemistry experiments, including protein purification, LigT reactions and TLC. D.B.-R. Performed all phylogenetics analyses and generated all phylogenetic trees. J.N. and N.P. performed cell-based luciferase assays. Y.K.Z., N.P. and J.N. generated Google Colab notebooks for data exploration. J.N. and J.A.D. wrote the original draft of the manuscript. All authors reviewed and edited the manuscript and support its conclusions.

**Competing interests** The Regents of the University of California have patents issued and pending for CRISPR technologies on which J.A.D. is an inventor. J.A.D. and J.N. are listed as

inventors on a patent filing related to DNA-binding proteins characterized in this work. J.A.D. is a cofounder of Azalea Therapeutics, Caribou Biosciences, Editas Medicine, Evercrisp, Scribe Therapeutics, Intellia Therapeutics and Mammoth Biosciences. J.A.D. is a scientific advisory board member at Evercrisp, Caribou Biosciences, Intellia Therapeutics, Scribe Therapeutics, Mammoth Biosciences, The Column Group and Inari. She also is an advisor for Aditum Bio. J.A.D. is Chief Science Advisor to Sixth Street, a Director at Johnson & Johnson, Altos and Tempus, and has research projects sponsored by Apple Tree Partners. The other authors declare no competing interests.

**Additional information**

**Correspondence and requests for materials** should be addressed to Jennifer A. Doudna.

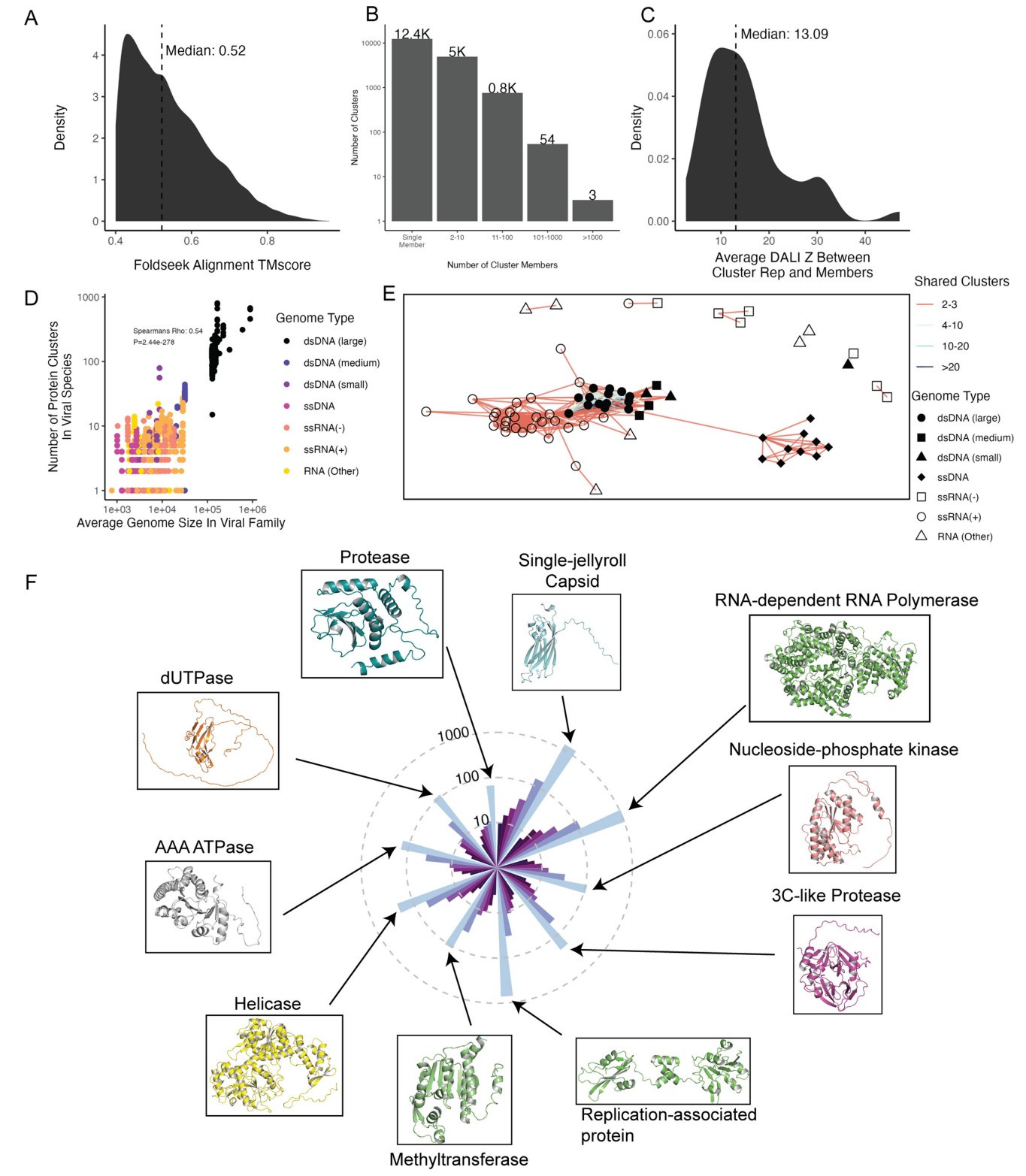

**Extended Data Fig. 1 |** See next page for caption.

**Extended Data Fig. 1 | Distribution of protein clusters across viral families.**
A. Foldseek was used to align all virus sequence cluster representatives against one another, and alignments with a TMscore below 0.4 were removed. This plot shows the distribution of alignment TMscores, with the X axis indicating the TMscore and the Y axis indicating the density (or "proportion") or alignments with each TMscore. B. The distribution of proteins amongst sequence clusters. The X axis indicates the size of each cluster, while the Y axis indicates the number of clusters of that size. C. For each protein cluster with at least 100 members, the cluster representative was aligned with DALI against all cluster members. Clusters that contained members with an average length of 150 residues or less were excluded, and members that did not align to the representative were assigned a Z score of 0. The distribution of average Z scores for each cluster is plotted, with the median cluster-averaged indicated. X axis indicates the DALI Z score for each cluster, while the Y axis indicates the density (or proportion) of clusters with each average DALI Z score. D. Relationship between the number of protein clusters encoded by a viral species (Y axis) and the average genome size of its family in nucleotides (X axis). Each dot is a viral species, and colors indicate the genome type. The spearman's (two-sided) Rho is 0.54, with a P value < 2.2e-16, indicating a strong correlation. E. Each node represents a single viral family, with the shape and color indicating the genome type of that family. The color of edges between the nodes indicates the number of shared protein clusters between each pair of families. Only those family-family pairs with at least 2 shared protein clusters are plotted. F. Protein clusters were ordered by their phylogenetic diversity of their members (e.g. # phyla > # classes > # orders >... # species) and the top 10 clusters were plotted. Bars are colored based and ordered on decreasing taxonomic level, with phyla as dark blue on the far left and species as bright blue on the far right of each stack.

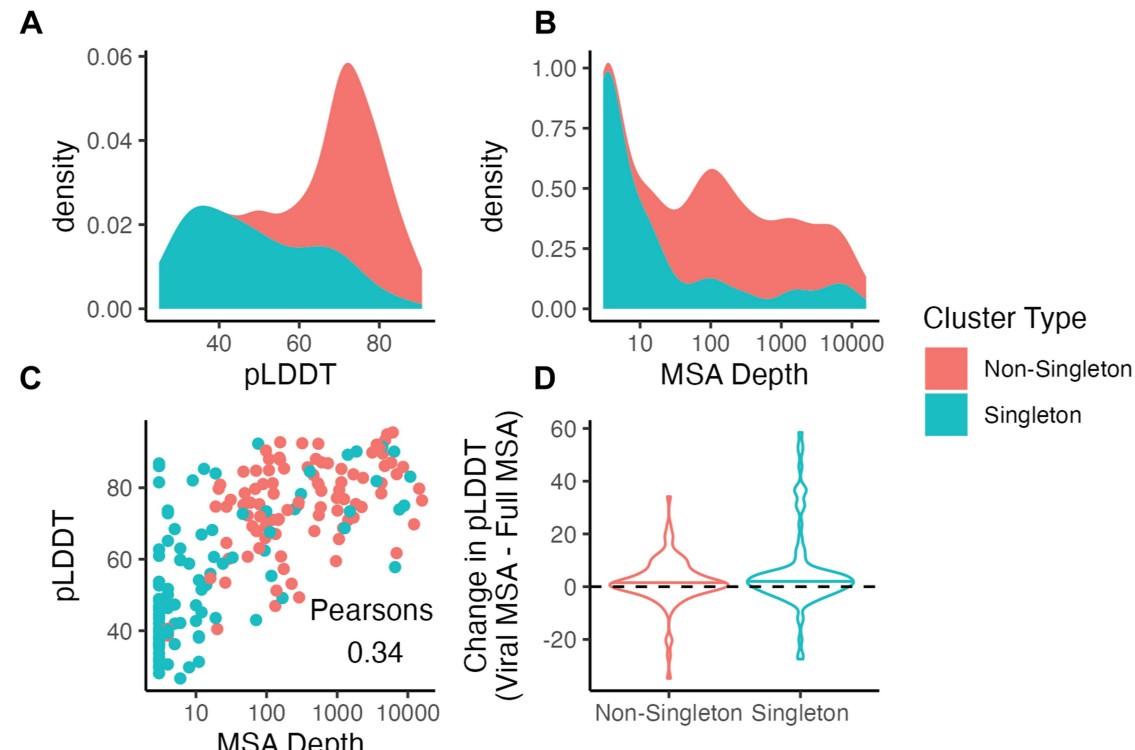

**Extended Data Fig. 2 | MSA generation against the full Colabfold MMseqs2 Database.** A. The protein representative for the top 100 protein clusters by size and from 100 random singleton clusters were selected, MSAs were generated against the full Colabfold MMSeqs2 database, and structures were predicted from this new MSA. The distribution of pLDDT values for structures from singleton (blue) or non-singleton (orange) clusters are plotted. The X axis indicates the pLDDT, while the Y axis indicates the density (or proportion) of proteins that have the indicated pLDDT value. B. The distribution of MSA depths is plotted for singleton (blue) and non-singleton (orange) clusters. The X axis indicates MSA depth and is log scale, while the Y axis indicates the density (or proportion) of proteins that have the indicated MSA depth. MSA depth is defined as the number of sequences in the MSA. C. For each protein, its pLDDT is plotted on the Y axis while its MSA depth is plotted on the X axis. Each dot is a protein, and the dots are colored according to whether they are from a singleton (blue) or non-singleton (orange) cluster. Pearsons (two-sided) correlation is 0.34 (95 percent confidence interval: 0.2137995, 0.4615760), P value 8.164e-07. D. For each of the 200 proteins studied, the average pLDDT of its structure created with the full Colabfold MSA is subtracted from its average pLDDT when folded with the viral MSA. This change is plotted on the Y axis, where a value above 0 indicates the viral MSA yielded a higher average pLDDT. The X axis indicates whether the proteins are from non-singleton or singleton clusters. The bars in each violin plot indicate the median of the plotted population.

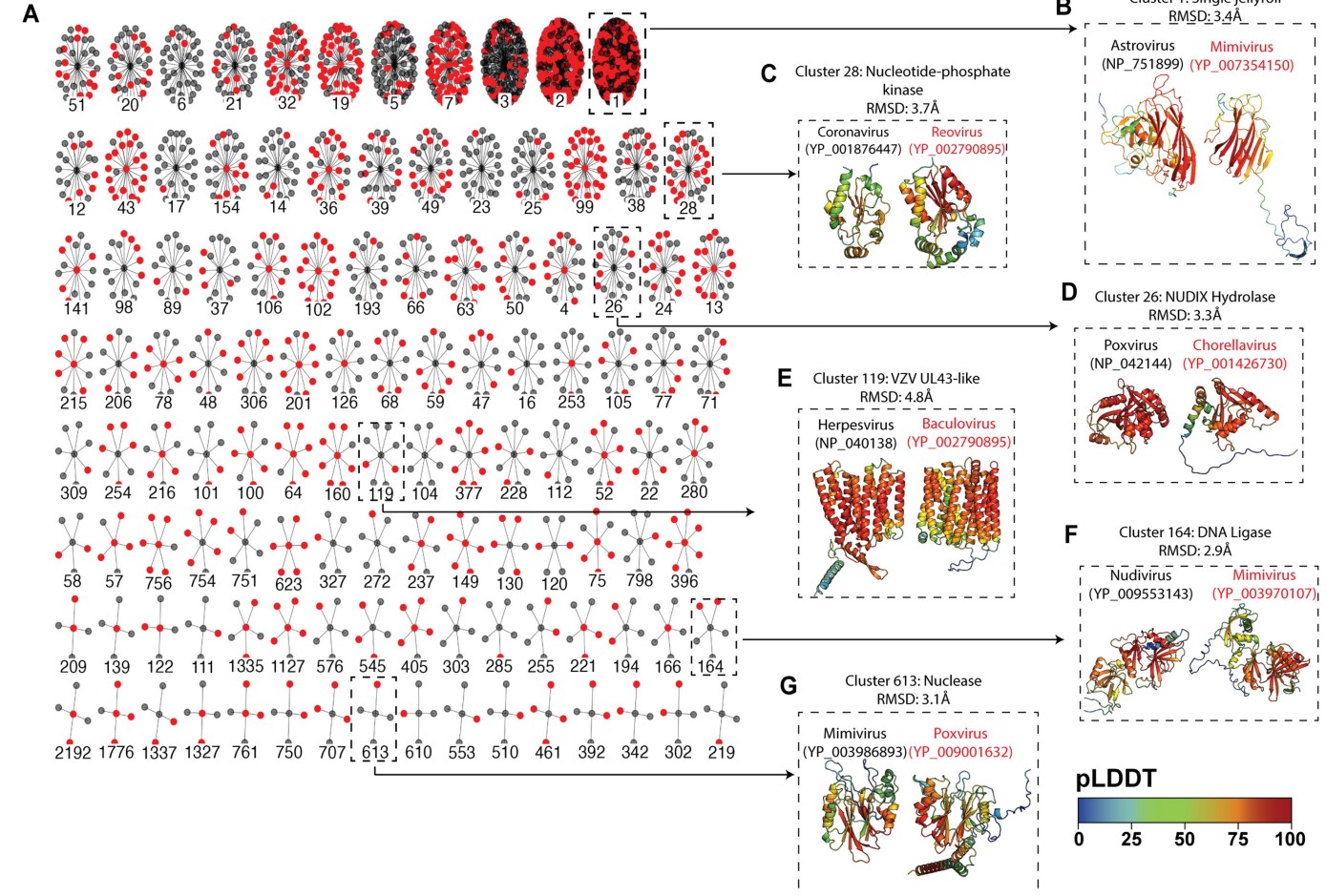

**Extended Data Fig. 3 | Many unannotated proteins have structural similarity to annotated protein clusters.** A. Many protein clusters contain a mix of annotated and unannotated sequence clusters. Each "wheel" of nodes indicates a protein cluster, with individual nodes representing individual sequence clusters. Each sequence cluster node is colored based on if it is annotated (gray) or unannotated (red). All protein clusters with at least one annotated and one unannotated protein cluster are shown. Numbers below each wheel indicate the cluster ID. B-G. (Left) A network of sequence clusters that belong to each protein cluster, where nodes that are red are unannotated and those that are gray are annotated. The centroid is the protein cluster representative. (Right) Members of annotated and unannotated sequence clusters are highlighted, where the structure of an annotated protein (left) is compared to the structure of an unannotated protein (right). Proteins are colored based on pLDDT, with red indicating higher pLDDT and blue indicating lower pLDDT. The RMSD between the two structures is indicated.

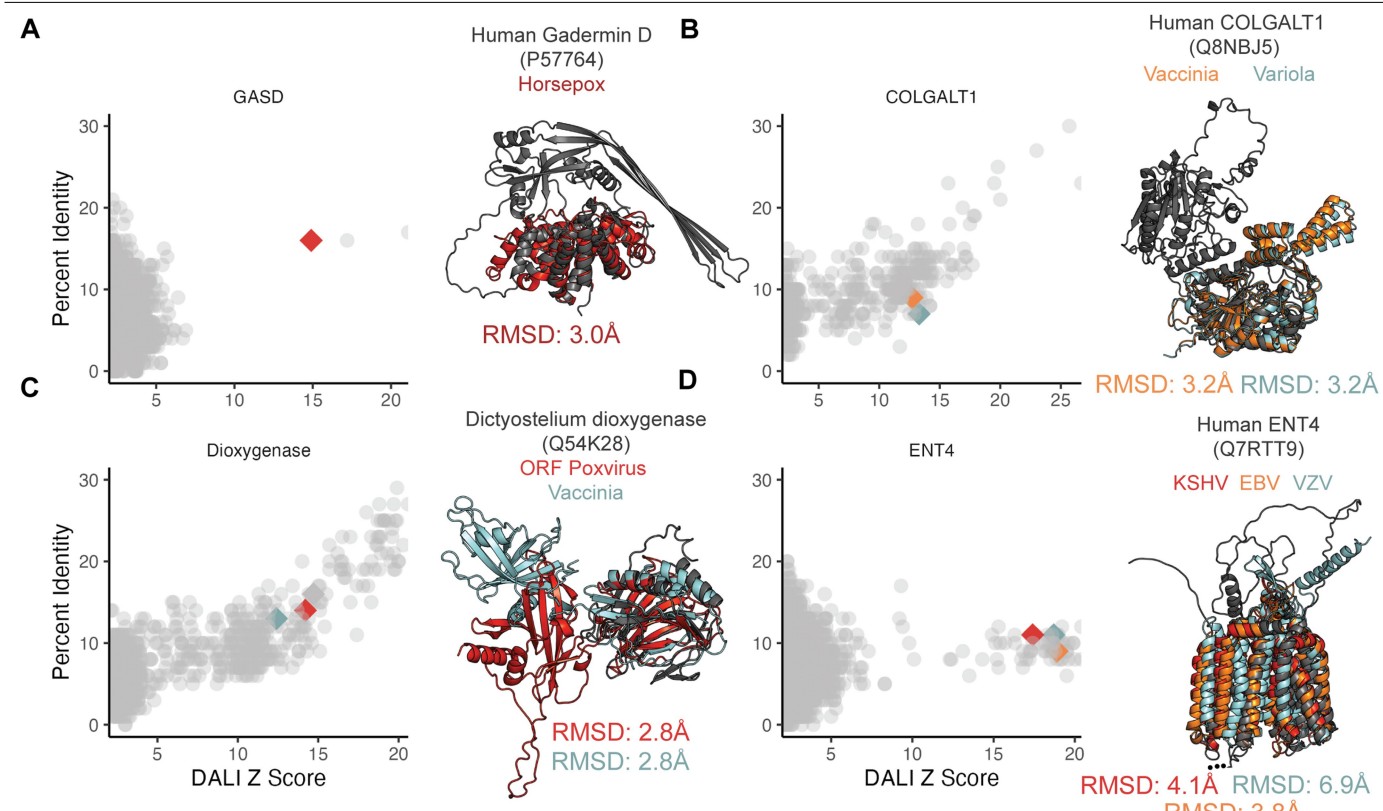

**Extended Data Fig. 4 | Structural similarities between viral and non-viral proteins.** A-D. Specific non-viral hits from the Alphafold Foldseek search were aligned against the viral predicted structure database using DaliLite, and alignments against proteins from human pathogens were selected. (Left) The Y axis indicates the percentage amino acid identity, and the X axis indicates the Dali Z score. Each dot indicates a single alignment. Each point indicates an alignment, with the points corresponding to the proteins highlighted on the right as diamonds and colored consistently with their protein structures. (Right) The structure of the non-viral protein query is present in black. A superposition of selected protein clusters is shown, with the RMSD of each viral protein vs the non-viral protein indicated. Protein accessions are as follows: GASD: (Horsepox-ABH08278). COLGALT1: (Vaccinia-YP_232983; Variola-NP_042130). Dioxygenase: (ORF Poxvirus-NP_957891; Vaccinia-YP_232906). ENT4: (KSHV-YP_001129415; VZV-NP_040138; EBV-YP_401658).

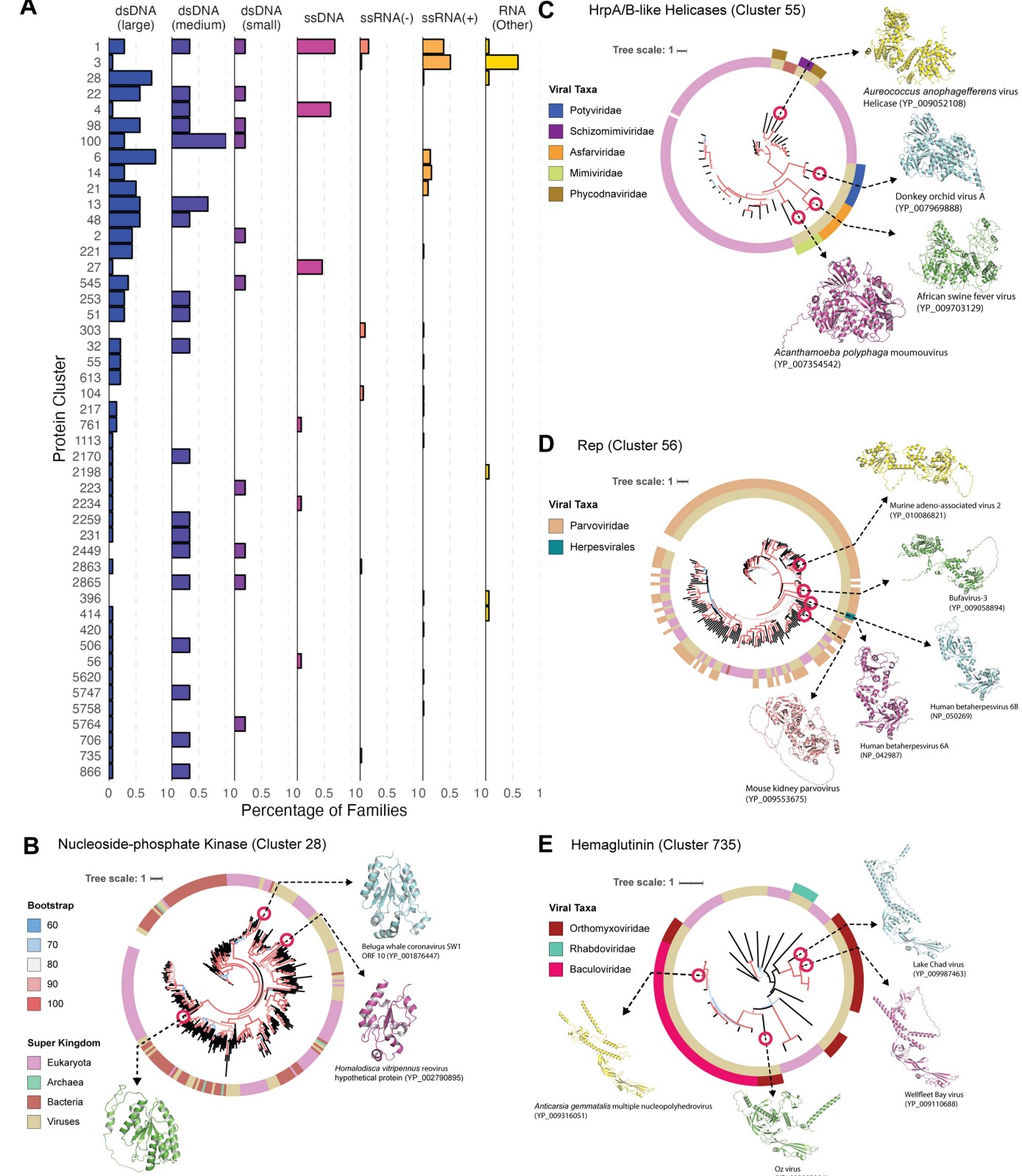

**Extended Data Fig. 5** | See next page for caption.

**Extended Data Fig. 5 | Horizontal gene transfer drives the emergence of taxonomically-diverse protein clusters.** A. Protein clusters were ranked as follows: 1) by the number of genome types of viral species that encode cluster members, followed by 2) the number of viral families that encode cluster members. The top 50 protein clusters by this metric were included in the plot. Each row is a protein cluster (with the number indicating the protein cluster ID). The X axis indicates the percentage of viral families of each genome type that contain a viral species that encodes a member of the protein cluster. B. A polyphyletic protein cluster of a nucleotide-phosphate kinase fold. The ring indicates the Superkingdom of each member of the tree. The structures of individual members are highlighted. The scale bar indicates substitutions per site. C. A polyphyletic protein cluster of HrpA/B-like helicases. The inner ring indicates the Superkingdom of each member of the tree, with the same color key as panel B. The outer ring indicates the viral taxa (here, viral family) of relevant members of the tree. The structures of individual members are highlighted. The scale bar indicates substitutions per site. D. A monophyletic protein cluster of Rep-like proteins shows sequence similarity between Parvovirus Rep proteins and a Rep-like protein in HHV6A and HHV6B. The inner ring indicates the Superkingdom of each member of the tree, with the same color key as panel B. The outer ring indicates the viral taxa (here, viral family) of relevant members of the tree. The structures of individual members are highlighted. The scale bar indicates substitutions per site. E. A monophyletic protein cluster of Hemagglutinin-like proteins shows sequence similarity between a clade of orthomyxovirus and baculovirus hemagglutinins. The inner ring indicates the Superkingdom of each member of the tree, with the same color key as panel B. The outer ring indicates the viral taxa (here, viral family) of relevant members of the tree. The structures of individual members are highlighted. The scale bar indicates substitutions per site.

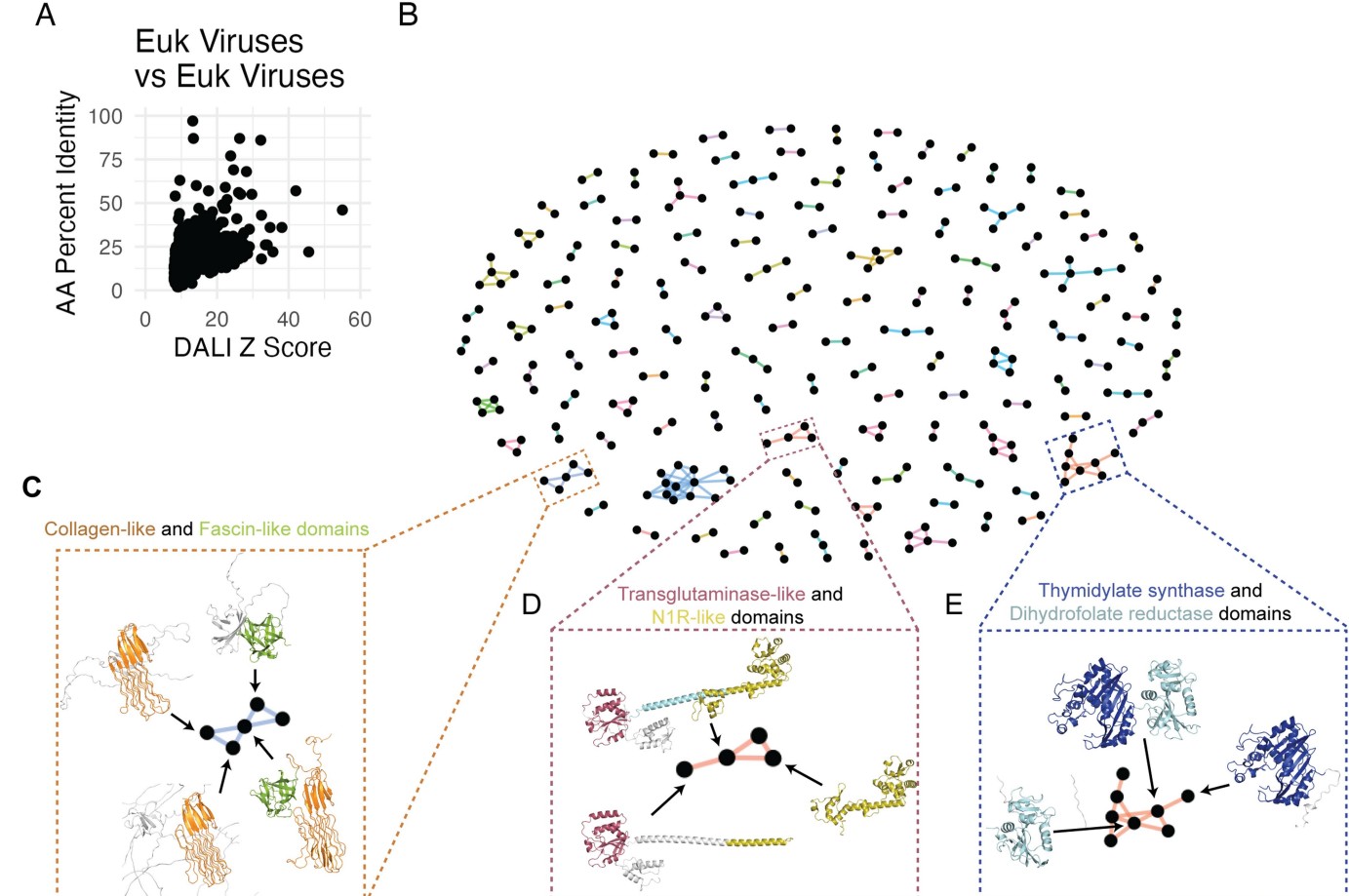

**Extended Data Fig. 6 | Shared domains across eukaryotic virus protein clusters.** A. All-by-all structural alignments of representative structures from the 5,770 protein clusters with more than one member. Each dot indicates a single alignment, with the Y axis indicating the fraction of amino acid identity and the X axis indicating DALI Z-score. B. Protein clusters tend to share protein domains. Each node indicates a protein cluster, and edges between protein clusters indicate there is a DALI alignment between them. Only alignments with a Z score of at least 15 are plotted. The boxes indicate cluster representatives highlighted in subsequent panels. C. Frequent reuse of structural/ cytoskeleton-related domains. Protein clusters with collagen-like domains (orange) and fascin-like domains (green) are highlighted. D. Multiple combinations of domains with the same viral genus. Diverse combinations of transglutaminase-like domains (purple) and N1R-like domains (green) from entomopoxvirus proteins are highlighted. E. Frequent reuse of protein domains involved in metabolism. Various combinations of thymidylate synthase (dark blue) and dihydrofolate reductase (light blue) domains in protein clusters are highlighted.

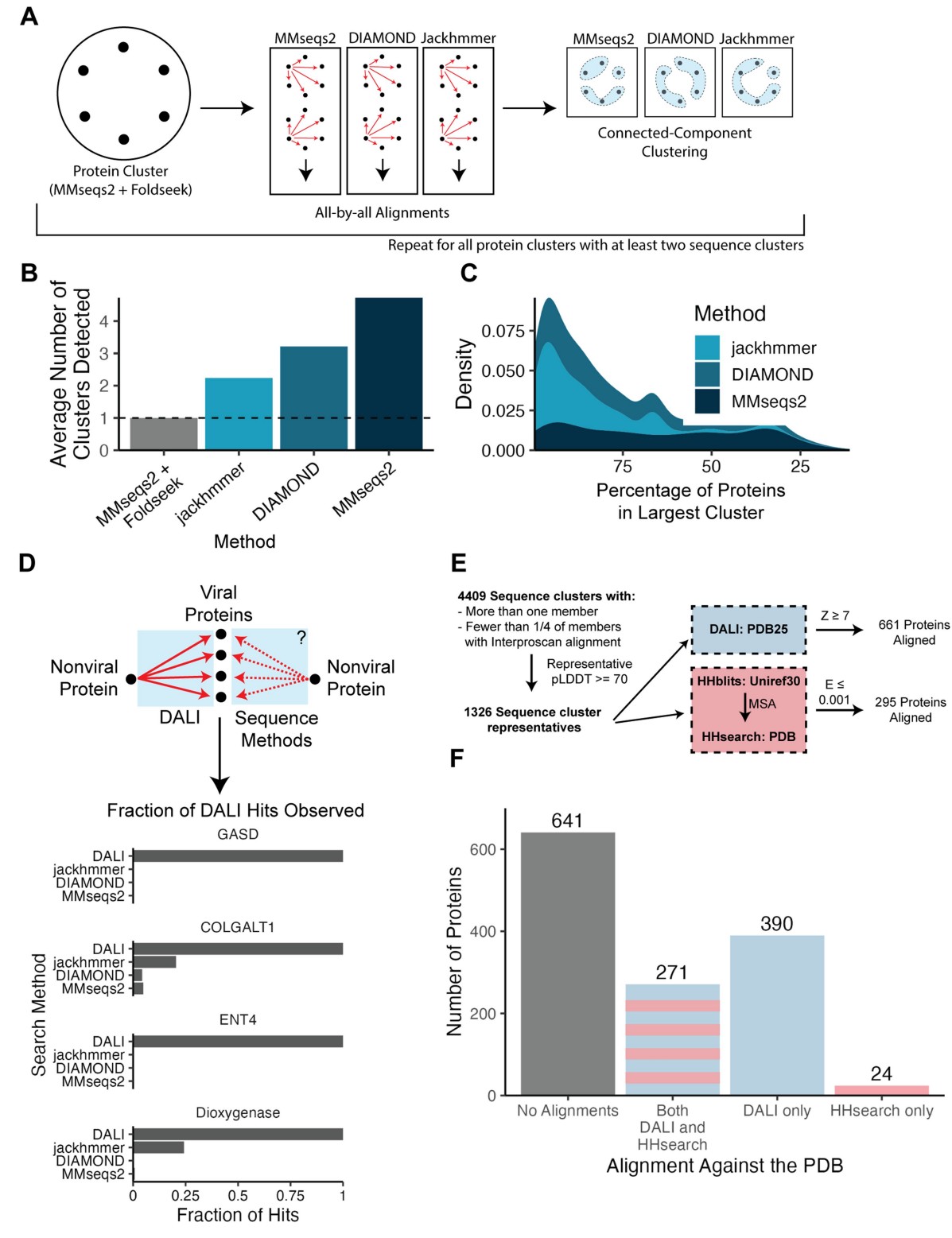

**Extended Data Fig. 7** | See next page for caption.

**Extended Data Fig. 7 | Structure methods outperform sequence methods at identifying virus-virus and host-virus protein similarities.** A. Method for doing benchmarking. For all protein clusters with at least two sequence clusters, we conducted all-by-all alignments between members using MMseqs2, DIAMOND blastp, and jackhmmer. These alignments and subsequence clustering occur separately for each protein cluster. From these alignments, we conducted connected-component clustering using sat.py aln_cluster. B. This plot indicates the average number of clusters detected (on the Y axis) by each method (on the X axis) across all of the protein clusters that contain at least two sequence clusters. C. For each sequence method, the proportion of original protein cluster members that were included in the largest cluster is plotted across all original protein clusters. The X axis indicates the proportion of proteins in the largest cluster for the indicated sequence method (color), while the Y axis indicates the density (or proportion) of original protein clusters with that value. D. To compare the sensitivity of structure and sequence alignment at detecting similarities between virus and non-virus proteins, we conducted sequence alignments using MMseqs2, DIAMOND, and jackhmmer to align each non-viral query against the viral database. These plots then indicate, for each query, the fraction of DALI alignments that are likewise identified through each sequence method. E. We identified swell-folded sequence cluster representatives from clusters containing no more than ¼ of members with an Interproscan alignment. We aligned these 1,326 proteins against the PDB using structural search (with DALI) or sequence search (with HHblits and HHsearch, similar to HHpred webserver). This resulted in 661 alignments with DALI and 295 alignments with HHblits/HHsearch. F. This bar plot indicates, for each of the 1,326 proteins in the benchmark set, the number of proteins with no alignments against the PDB with either DALI or HHblits/HHsearch, alignments against the PDB with both DALI and HHblits/HHsearch, or alignment against the PDB with only DALI or HHblits/HHsearch.

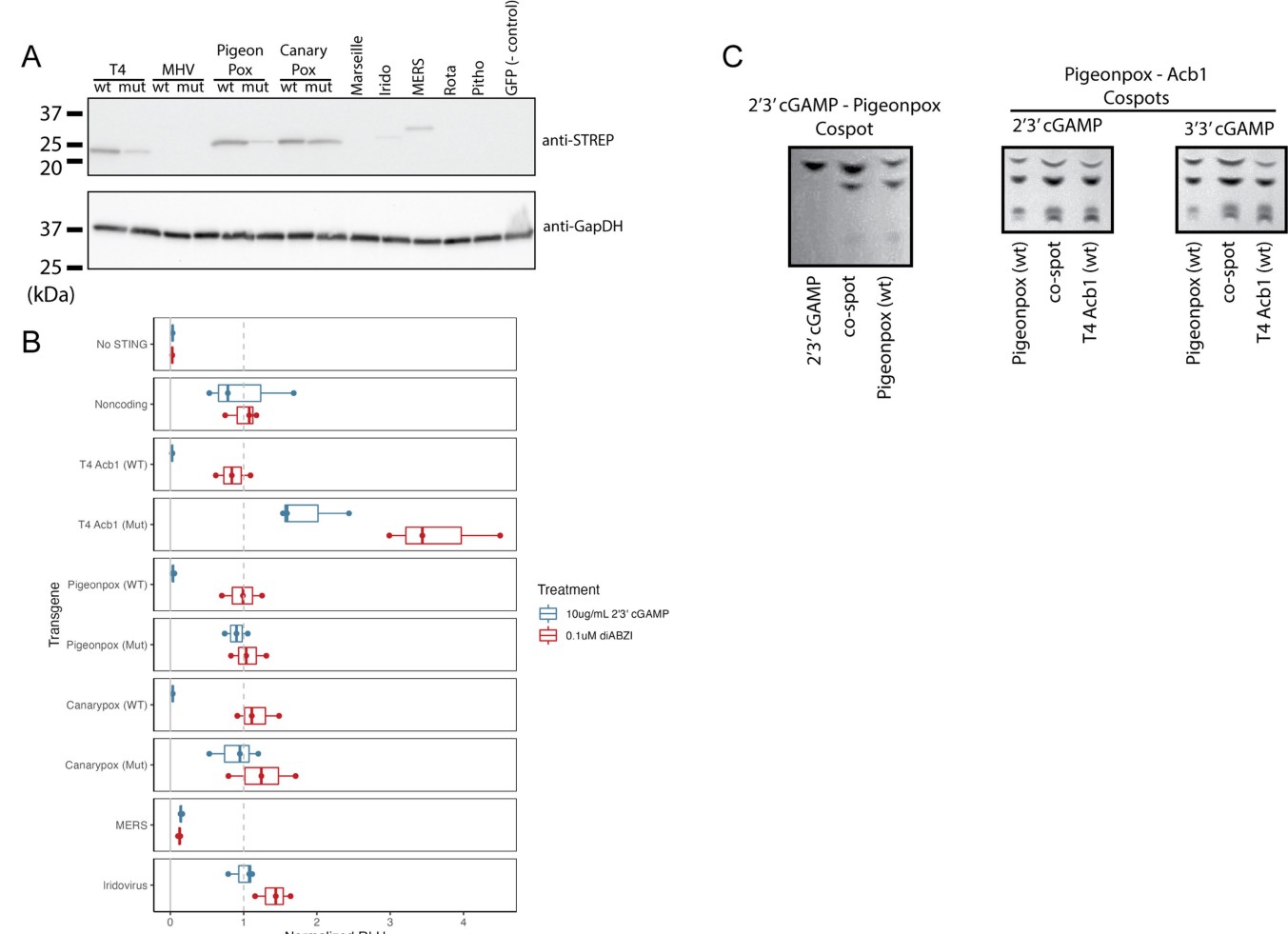

**Extended Data Fig. 8 | Activity of viral PDEs against 2'3' cGAMP.** A. Western blot of 293 T cells following transfection of each viral LigT-like PDE. Each LigT-like PDE contains two STEP2 tags, and were visualized using an anti-STREP antibody. This blot is representative of two independent experiments. For gel source data see Supplementary Fig. 1. B. This experiment was conducted as illustrated in Fig. 4c. The X axis indicates the relative RLU normalized to the Noncoding transgene condition for each STING agonist (either 2'3' cGAMP or diABZI)."Mut" indicates the transgene contains mutations of both catalytic histidines. Boxplots indicate 25th, 50th, and 75th percentiles, while whiskers go to the highest or smallest point up to 1.5 * interquartile range. Plotted data are from one biological replicate and three wells per condition. C. Thin-layer chromatography of co-spots between different conditions. (Left) Co-spotting of 2'3' cGAMP and the Pigeonpox/2'3'cGAMP reaction. (Middle) Co-spotting of the Pigeonpox (WT)/2'3' cGAMP reaction and the Acb1 (WT)/2'3' cGAMP reaction. (Right) Co-spotting of the Pigeonpox (WT)/3'3' cGAMP reaction and the Acb1 (WT)/3'3' cGAMP reaction.

# Reporting Summary

## Statistics

For all statistical analyses, confirm that the following items are present in the figure legend, table legend, main text, or Methods section.

| n/a | Confirmed | |
|---|---|---|
| ☐ | ☒ | The exact sample size (*n*) for each experimental group/condition, given as a discrete number and unit of measurement |
| ☐ | ☒ | A statement on whether measurements were taken from distinct samples or whether the same sample was measured repeatedly |
| ☐ | ☒ | The statistical test(s) used AND whether they are one- or two-sided<br>*Only common tests should be described solely by name; describe more complex techniques in the Methods section.* |
| ☒ | ☐ | A description of all covariates tested |
| ☒ | ☐ | A description of any assumptions or corrections, such as tests of normality and adjustment for multiple comparisons |
| ☐ | ☒ | A full description of the statistical parameters including central tendency (e.g. means) or other basic estimates (e.g. regression coefficient) AND variation (e.g. standard deviation) or associated estimates of uncertainty (e.g. confidence intervals) |
| ☐ | ☒ | For null hypothesis testing, the test statistic (e.g. *F*, *t*, *r*) with confidence intervals, effect sizes, degrees of freedom and *P* value noted<br>*Give P values as exact values whenever suitable.* |
| ☒ | ☐ | For Bayesian analysis, information on the choice of priors and Markov chain Monte Carlo settings |
| ☒ | ☐ | For hierarchical and complex designs, identification of the appropriate level for tests and full reporting of outcomes |
| ☐ | ☒ | Estimates of effect sizes (e.g. Cohen's *d*, Pearson's *r*), indicating how they were calculated |

*Our web collection on statistics for biologists contains articles on many of the points above.*

## Software and code

Policy information about availability of computer code

| Data collection | Code used for upstream processing is present in the vpSAT Github repository (https://github.com/jnoms/vpSAT/tree/main). This includes scripts required for most computational steps. A workflow is available that shows all main processing steps: https://github.com/jnoms/vpSAT/blob/main/manuscript_code/latest/analysis_workflow.ipynb. The stable vpSAT version used for this work is available through zenodo: https://zenodo.org/doi/10.5281/zenodo.10373132.<br><br>The SAT python package can be downloaded as instructed on the SAT Github repository (https://github.com/jnoms/SAT/tree/main). The stable SAT version used for this work is available through zenodo: https://zenodo.org/doi/10.5281/zenodo.10373132.<br><br>Code used for phylogenetics analysis can be found here: https://github.com/Doudna-lab/nomburg_j-LigT_phylogeny.<br><br>All plotting and analysis scripts are available as Quarto documents: https://github.com/jnoms/vpSAT/blob/main/manuscript_code/2024-01-04/.<br><br>All code can also be found on Zenodo, along with all intermediate data necessary to reproduce the figures: https://zenodo.org/doi/10.5281/zenodo.10373132<br><br>Other software versions:<br>MMseqs2 release version b0b8e85f3b8437c10a666e3ea35c78c0ad0d7ec2<br>Colabfold downloaded June 22, 2023<br>Dalilite version 5<br>Foldseek version 2.8bd520 |
|---|---|

IQTree version 2.3.3
Clustal Omega v1.2.4
MMseqs2 version 15.6f452 (for the phylogenetics analysis)
InterProScan version 5
PSI-BLAST version 2.15.0
DIAMOND version 0.9.14
jackhmmer version 3.1b2
hhsuite version 3.3.0

| Data analysis | R version 4.0.3 - all analysis code is available in the Zenodo repository |

For manuscripts utilizing custom algorithms or software that are central to the research but not yet described in published literature, software must be made available to editors and reviewers. We strongly encourage code deposition in a community repository (e.g. GitHub). See the Nature Portfolio guidelines for submitting code & software for further information.

## Data

Policy information about availability of data

All manuscripts must include a data availability statement. This statement should provide the following information, where applicable:
- Accession codes, unique identifiers, or web links for publicly available datasets
- A description of any restrictions on data availability
- For clinical datasets or third party data, please ensure that the statement adheres to our policy

There are several options for viewing, downloading, and searching the structural models generated here.

Searching
We have established a Google Colab notebook that enables any user to quickly and easily search one or more protein structures against our viral structure database using Foldseek: https://colab.research.google.com/github/jnoms/vpSAT/blob/main/bin/colab/QueryStructures.ipynb . This notebook runs rapidly and displays alignment results and information on the protein clusters to which alignment targets belong.
For users who want to conduct high-throughput searches, we have released a pre-made Foldseek database to facilitate use - https://zenodo.org/doi/10.5281/zenodo.10685504.

Viewing and downloading
We have established a Google Colab notebook that allows users to explore our data. Users can input a virus taxonomy ID or family name and browse available proteins. Users can then automatically view and download individual structures. https://colab.research.google.com/github/jnoms/vpSAT/blob/main/bin/colab/ExploreStructures.ipynb .
We have uploaded our structures to ModelArchive - https://www.modelarchive.org/doi/10.5452/ma-jd-viral. ModelArchive hosts predicted structures in a uniform way with extensive metadata. Furthermore, ModelArchive is part of EBI's 3D-Beacons framework (https://www.ebi.ac.uk/pdbe/pdbe-kb/3dbeacons/), enabling uniform downloads and processing of our protein structures through a shared API encompassing the PDB, Alphafold database, and other databases.
Structures can be accessed through each viral family phage in Viralzone [ref] - https://viralzone.expasy.org/10977.
Finally, all structures are available on Zenodo: https://zenodo.org/doi/10.5281/zenodo.10291580

## Research involving human participants, their data, or biological material

Policy information about studies with human participants or human data. See also policy information about sex, gender (identity/presentation), and sexual orientation and race, ethnicity and racism.

| Reporting on sex and gender | N/A |
| Reporting on race, ethnicity, or other socially relevant groupings | N/A |
| Population characteristics | N/A |
| Recruitment | N/A |
| Ethics oversight | N/A |

Note that full information on the approval of the study protocol must also be provided in the manuscript.

# Field-specific reporting

Please select the one below that is the best fit for your research. If you are not sure, read the appropriate sections before making your selection.

☒ Life sciences          ☐ Behavioural & social sciences          ☐ Ecological, evolutionary & environmental sciences

For a reference copy of the document with all sections, see nature.com/documents/nr-reporting-summary-flat.pdf

# Life sciences study design

All studies must disclose on these points even when the disclosure is negative.

| | |
|---|---|
| Sample size | There was no consideration of sample size. We simply included all viral proteins available from RefSeq at the start of this study. |
| Data exclusions | There were two key exclusions:<br>1. Proteins that were larger than 1500 residues (or, in some cases 1000 residues) were excluded. This resulted in exclusion of 1706 proteins.<br>2. A small number of proteins failed to fold. |
| Replication | All computational pipelines were run successfully at least twice. |
| Randomization | This study did not perform experiments that require sample randomization. |
| Blinding | This study did not perform experiments that require blinding. |

# Reporting for specific materials, systems and methods

We require information from authors about some types of materials, experimental systems and methods used in many studies. Here, indicate whether each material, system or method listed is relevant to your study. If you are not sure if a list item applies to your research, read the appropriate section before selecting a response.

## Materials & experimental systems

| n/a | Involved in the study |
|---|---|
| ☐ | ☒ Antibodies |
| ☐ | ☒ Eukaryotic cell lines |
| ☒ | ☐ Palaeontology and archaeology |
| ☒ | ☐ Animals and other organisms |
| ☒ | ☐ Clinical data |
| ☒ | ☐ Dual use research of concern |
| ☒ | ☐ Plants |

## Methods

| n/a | Involved in the study |
|---|---|
| ☒ | ☐ ChIP-seq |
| ☒ | ☐ Flow cytometry |
| ☒ | ☐ MRI-based neuroimaging |

## Antibodies

| | |
|---|---|
| Antibodies used | Streptactin HRP (IBA 2-1502-001), Santa Cruz Biotech Mouse anti GapDH (sc-365062), ECL Anti-mouse IgG (Amersham NXA931) |
| Validation | These antibodies are reported to be validated by their manufacturers. |

## Eukaryotic cell lines

Policy information about cell lines and Sex and Gender in Research

| | |
|---|---|
| Cell line source(s) | HEK 293T cells were kindly provided by the Ott lab at the Gladstone Institutes. These cells were originally provided by ATCC. |
| Authentication | None of the cell lines were authenticated. |
| Mycoplasma contamination | Cells were tested for Mycoplasma by the Gladstone Institutes Stem Cell Core within the past year, and determined to be Mycoplasma free. |
| Commonly misidentified lines<br>(See ICLAC register) | No commonly misidentified cell lines were used in this study. |

## Plants

Seed stocks

*Report on the source of all seed stocks or other plant material used. If applicable, state the seed stock centre and catalogue number. If plant specimens were collected from the field, describe the collection location, date and sampling procedures.*

Novel plant genotypes

*Describe the methods by which all novel plant genotypes were produced. This includes those generated by transgenic approaches, gene editing, chemical/radiation-based mutagenesis and hybridization. For transgenic lines, describe the transformation method, the number of independent lines analyzed and the generation upon which experiments were performed. For gene-edited lines, describe the editor used, the endogenous sequence targeted for editing, the targeting guide RNA sequence (if applicable) and how the editor was applied.*

Authentication

*Describe any authentication procedures for each seed stock used or novel genotype generated. Describe any experiments used to assess the effect of a mutation and, where applicable, how potential secondary effects (e.g. second site T-DNA insertions, mosiacism, off-target gene editing) were examined.*

