## [Peer Review File · Nature]

Manuscript Title: Birth of protein folds and functions in the virome

Reviewer Comments & Author Rebuttals

Reviewer Reports on the Initial Version:

Referees' comments:

Referee #1 (Remarks to the Author):

Nomburg et al report a new database of >60,000 eukaryotic viral protein structure predictions, global phylogenetic analysis of these proteins and potential horizontal transfer events, and identification of potential connections between viral and host proteins. The results build on recent advances enabling accurate protein structure prediction (AlphaFold2/ColabFold) and large-scale analysis of protein structural homology (FoldSeek). Major highlights of the research include establishing a database of eukaryotic viral structure predictions (a notable limitation in currently available datasets) and identifying a potential cyclic dinucleotide-targeting enzyme in avian poxviruses. The impact of the research could be greatly improved in two ways: (1) ensuring access and analysis of the new database is widely available to researchers with limited bioinformatics expertise, and (2) expanding experimental analysis of new viral protein functions.

Major comments:

1) The authors' new database of >60,000 eukaryotic viral protein structure predictions has the potential to be an incredibly transformative advance. Maximizing use of this resource will require greater data and method accessibility than is currently possible from the paper methods and the proposed Zenodo directory. Ideally, can the authors create a web interface or tool that allows any user to search based on a protein sequence or protein coordinate file to identify structurally related viral homologs and specific clusters determined in the authors' analysis?

ColabFold and FoldSeek continue to have a transformative impact on many fields in part because of the incredible accessibility of these tools to all scientists. Without creating similar accessibility options (or perhaps interfacing directly with FoldSeek) it is unclear if the authors' work will be as impactful for the research community.

2) The authors present several new interesting hypotheses about the function of viral protein clusters. In several cases these observations already exist in the literature (poxvirus gasdermin and galactosyltransferase homologs) limiting demonstration of the impact of the authors' new large-scale structural database. Can the authors highlight further examples of connections from their database or provide further experimental data to test the novel hypotheses related to viral transporter-like or dioxygenase-like protein function?

3) The authors' data defining avian poxvirus LigT-family phosphodiester enzymes as inhibitors of cGAS-STING immunity is incomplete to support claims in the paper.

3a. The claims "enzymatic activity validated in cell based experiments" and "a novel mechanism of 2'3' cGAMP degradation by eukaryotic viruses" are not fully proven by the data. What is the evidence that LigT-like proteins are enzymes? Could these proteins instead bind and sequester 2'3'-cGAMP or prevent STING activation through an indirect mechanism?

3b. It is confusing why experiments are not performed to directly test the ability of viral LigT-like proteins to degrade 2'3'-cGAMP and related nucleotide immune signals. The authors are world-leading experts in biochemistry and several readily available methods exist that could be used to measure the ability of LigT-like proteins to degrade 2'3'-cGAMP. For example, the authors could test viral LigT-like proteins using recombinant protein and commercially available 2'3'-cGAMP in degradation analysis by TLC, HPLC, or ion-exchange chromatography. Demonstrating that viral LigT-like proteins use a similar mechanism to degrade 2'3'-cGAMP as phage Acb1 proteins would help verify the proposed structural connection.

Minor comments:

4) The title of the paper includes "birth of new protein folds" but new protein folds are not described or verified. What is the evidence of new protein folds (compared to proteins that simply do not have robust matches in FoldSeek or DALI analysis)?

5) It is unclear what the authors mean by describing protein folds as "evolutionarily young"? Are all predicted structures that have rare matches necessarily recently evolved?

6) In structural alignments between annotated and unannotated sequence clusters (Figure 2), can the authors additionally show the structure of the unannotated sequence colored by LDDT? Are high confidence residues driving the alignment score?

7) In the clustering analysis, 20% (>12,000) of the total viral sequences cluster as singletons (Figure S1B). Can the authors comment further on what this means? Is this due to limited MSAs, low confidence structure predictions, or possible correlations with multi-domain proteins or disordered regions?

8) In Supplemental Figure 2 It is unclear what the numbers beneath each "wheel" mean. Is this simply a cluster ID?

9) Description of poxins 2'3'-cGAMP nucleases should cite the original primary work of Eaglesham et al 2019 (PMID 30728498).

I hope the authors will find my comments useful, thank you for the opportunity to read this interesting manuscript.

Philip Kranzusch (with Sam Fernandez and Sam Hobbs)

Referee #1 (Remarks on code availability):

The database and code do not appear to be available on Zenodo. This is a major point in my review, the accessibility of these data is a key part of the impact of the research.

Referee #2 (Remarks to the Author):

The manuscript by Nomburg et al. describes an extensive sequence and structural analysis of the proteomes of 4,463 viral species across 132 different viral families. The clustering analysis recapitulates many evolutionary tenets: proteins central to the viral cycle are predominant, structure conservation is far more extent than sequence conservation. One interesting point is the finding of many clusters with significant structural homology with clusters of other viral families, as well as non-viral clusters, suggesting the presence of horizontal gene transfer. The most important contribution of the paper appears the existence of two acquisition events within the betacoronavirus genus, of PDEs, proteins involved in the evasion of the oligoadenylate synthase pathway, a highly conserved immune mechanism against double-stranded viruses.

In my opinion, the manuscript is rigorous, polished and technically excellent, it is well-written and it addresses interesting questions in fundamental evolutionary biology. The analysis is clear and intuitive, and the scale of the computational work is impressive. In my opinion, the article should definitely be published in some form, and I would like to congratulate the authors for their excellent work.

My key concern with this manuscript — and I am sorry that it is something so non-scientific — is whether the scope of the novelty warrants its publication in a top journal like Nature. The main conclusion appears to be the discovery of the genetic signatures of the formation of the pathway. While I am not a virologist and therefore cannot fully comment on the relevance of this finding, it does not appear sufficiently novel to warrant publication. From the field of structural bioinformatics — which is my field of expertise, and why I suspect I was asked to review this paper —, although the authors' methodology is sound and the scale of application is impressive, I do not find their methods particularly novel. Given that the authors also only seem to find one new piece of information, I am not convinced that the methodology provides a general method for hypothesis generation and/or testing to characterise viral protein evolution and function across the virome.

I also have a few minor comments to provide to the authors:

- The authors generate protein structure predictions using ColabFold, although with manually generated MSAs to enhance performance. The alignments are generated using MMseqs2 on the RefSeq virus database. While I understand the authors' choice, some publications, if not popular folklore around AlphaFold usage, suggests that incorporating metagenomics sequences into the alignment is crucial for

getting good predictions in rapidly evolving proteins. I was wondering if the authors would be able to provide an empirical validation of the effect of incorporating all databases to the alignment (i.e. UniRef + BFD + Mgnify), even if only for a few representative proteins. This test would have to be done not only on the larger clusters (where there are obviously many homologs) but also on some of the ~60% clusters that have a single representative.

- The authors perform clustering at the sequence level and then use different structural techniques to understand shared evolutionary relations, which is illuminating. Nevertheless, there is one test that I was missing: if one runs structural searches for the ~12,000 clusters that have only one representative against the entire ~60k structures, is it possible to find structural homologs that were previously ignored? I would first test this using Foldseek, although intuition suggests (because the structure predictions may be of lower quality for these isolated clusters due to the lack of homologs) that DALI's increased sensitivity will be necessary for this experiment.

In accordance with my comments above, my recommendation to the editor would be that the manuscript is transferred to another journal of the Nature family (e.g. Nature Communications) without further review, where my recommendation would be of acceptance after addressing the minor comments posed above.

Referee #3 (Remarks to the Author):

Romburg et al used modern bioinformatic tools such as Colabfold to predict the structures of more than 60,000 proteins of eukaryotic viruses that have been sequenced. They found that 62% of these viral proteins do not have homologs in the AlphaFold database. For those viral proteins that have homologs in the AlphaFold database, putative functions of these viral proteins can be inferred based on homology to proteins with known functions. The authors focused on a class of proteins that contain the RNA ligase T (LigT)-like phosphodiesterase (PDE) domain, which has been found in the bacteriophage protein Acb1, an enzyme known to degrade the second messenger molecule cGAMP. They tested some of these putative enzymes and found that the LigT-like PDEs from pigeon and canary could indeed specifically inhibit STING activation by 2'3'-cGAMP.

The structural prediction of eukaryotic viral proteins provides a very useful resource for the community to study the functions of many viral proteins including those that have not been annotated. The authors validated their approach by providing an example of a class of viral enzymes that degrade cGAMP. However, the characterization of these enzymes is quite preliminary, especially considering that another class of cGAMP PDE has already been reported in poxviruses (known as poxin; PMID 30728498). This reviewer suggests that the authors further characterize the LigT family of enzymes as outlined below:

- 1) The reporter assay shown in Figure 4E does not prove that the avian poxvirus proteins cleave cGAMP; the results only showed that the proteins inhibited STING activation by cGAMP. In fact, several of the viral proteins (WT and/or mutant) also inhibited STING activation by diABZi, which raises the possibility

that some of these proteins might interfere with STING function through other mechanisms. The authors should carry out a biochemical assay to test directly whether the viral proteins indeed degrade cGAMP in vitro and if so, what are the degradation products. In Figure 4E and Figure S5A, they should also show Western blots to evaluate the protein expression levels in order to draw a valid conclusion.

2) The authors claimed that several of the ligT-like proteins from RNA viruses could cleave 2-5A but did not show any evidence. They should do the experiments to test the activity of these enzymes against 2-5A or other nucleotide messengers.

3) Figure S5A shows that the ligT-like protein from MERS (NS4b) could also inhibit STING activation by cGAMP but did not do the control experiments such as diABZi stimulation or using a catalytic mutant. Considering that MERS is a human pathogen, performing these experiments can enhance the paper. Do SARS viruses also encode such ligT-like enzymes?

4) The authors should try to characterize several ligT-like proteins shown in Figure 4B for their enzymatic activity and substrate specificity (e.g, specificity to various cyclic dinucleotides). This is important for establishing this family of proteins as viral PDEs that antagonize host immune defense.

Referee #4 (Remarks to the Author):

The manuscript by Nomburg, Price and Doudna describes comprehensive modeling a comparative structural analysis of viral proteins using AlphaFold2, FoldSeek and DALI. In addition, the authors experimentally test one of the resulting predictions, cGAMP like activity of viral PDEs that share the LigT fold.

AlphaFold2 is a breakthrough in protein structure and evolution analysis, but predictions of viral protein structures are conspicuously missing in the currently available database of AlphaFold predictions. The authors of this paper fill this gap and thus create a useful resource that will be appreciated and utilized by many researchers.

However, in the view of this reviewer, this is all that is achieved in this work, at least, as far as the computational part is conserved. The authors repeatedly claim novelty in terms of prediction of protein structures, evolutionary relationships and functions. However, the actual extent of novelty is impossible to assess, above all, because there is no effort to compare the performance of the structural analysis with that of sequence based methods, but also because some of the most relevant recent publications are neglected. A characteristic example is the "discovery" of the poxvirus gasdermin homologs. The authors identify this relationship by structural comparison and cite a preprint by Elde and colleagues reporting the same. Incidentally, citing the preprint is misleading because a substantially updated version of the cited work has since been published:

<https://pubmed.ncbi.nlm.nih.gov/37494187/>

More importantly, however, minimal attention to the published literature would have revealed that the same discovery has been reported based solely on sequence comparisons, and the evolutionary history of gasdermins in poxviruses has been traced in detail: <https://pubmed.ncbi.nlm.nih.gov/34253028/>.

This example shows that without systematic benchmarking, the impact of structural analysis remains uncertain. Even in cases where structural comparison is indeed important for discovering relationships between proteins, the authors overlook preceding publications that happen to report a far deeper analysis. A case in point is the poxvirus C1 protein for which the authors report the dioxygenase fold. In a previous publication, AlphaFold2/DALI were employed to identify this fold not only in C1 but in several other poxvirus proteins, and importantly, the patterns of inactivation of the catalytic sites of this enzyme were explored:

<https://pubmed.ncbi.nlm.nih.gov/37017580/>. This case is particularly suspicious because Nomburg et al only report one poxvirus dioxygenase homolog.

Even when it comes to the "crown jewel" of this work, the LigT-like PDEs, the degree of novelty is unclear given this recent publication:

<https://pubmed.ncbi.nlm.nih.gov/38277437/>

(also cited as a preprint although in this case, article was not yet published at submission of Nomburg et al.)

In summary, without careful benchmarking and incorporation (and proper citation) of previously published work, the analysis presented in this manuscript has virtually no merit.

The manuscript by Nomburg et al often refers to evolutionary events such as horizontal gene transfer, but the analysis leading to conclusions on such events is rudimentary and methodologically faulty. For instance, the authors use BLAST scores to infer horizontal gene transfer which is strictly not done in evolutionary biology. Hand in hand with this methodological inadequacy comes inappropriate language such "sequence homology" and structural homology" that appear throughout the manuscript.

Referee #4 (Remarks on code availability):

The code used in this paper is quite straightforward and does not require review

Author Rebuttals to Initial Comments:

Referee #1

Nomburg et al report a new database of >60,000 eukaryotic viral protein structure predictions, global phylogenetic analysis of these proteins and potential horizontal transfer events, and identification of potential connections between viral and host proteins. The results build on recent advances enabling accurate protein structure prediction (AlphaFold2/ColabFold) and large-scale analysis of protein structural homology (FoldSeek). Major highlights of the research include establishing a database of eukaryotic viral structure predictions (a notable limitation in currently available datasets) and identifying a potential cyclic dinucleotide-targeting enzyme in avian poxviruses. The impact of the research could be greatly improved in two ways: (1) ensuring access and analysis of the new database is widely available to researchers with limited bioinformatics expertise, and (2) expanding experimental analysis of new viral protein functions.

Major comments:

1) The authors' new database of >60,000 eukaryotic viral protein structure predictions has the potential to be an incredibly transformative advance. Maximizing use of this resource will require greater data and method accessibility than is currently possible from the paper methods and the proposed Zenodo directory. Ideally, can the authors create a web interface or tool that allows any user to search based on a protein sequence or protein coordinate file to identify structurally related viral homologs and specific clusters determined in the authors' analysis?

ColabFold and FoldSeek continue to have a transformative impact on many fields in part because of the incredible accessibility of these tools to all scientists. Without creating similar accessibility options (or perhaps interfacing directly with FoldSeek) it is unclear if the authors' work will be as impactful for the research community.

Response: Thank you for the great suggestion. We have taken several steps to make our database more accessible. We separate these steps into two categories: Searching and Viewing.

Searching:

1. We have established a Google Colab notebook that enables any user to quickly and easily search one or more protein structures against our viral structure database using Foldseek - <https://colab.research.google.com/github/jnomS/vpSAT/blob/main/bin/colab/QueryStructures.ipynb> . This notebook runs rapidly and displays alignment results and information on the protein clusters to which alignment targets belong.
2. For users who want to conduct high-throughput searches, we have released a pre-made Foldseek database to facilitate use - <https://zenodo.org/doi/10.5281/zenodo.10685504>

Viewing:

1. We have established a Google Colab notebook that allows users to explore our data. Users can input a virus taxonomy ID or family name and browse available proteins. Users can then automatically view and download individual structures. <https://colab.research.google.com/github/jnoms/vpSAT/blob/main/bin/colab/ExploreStructures.ipynb>
2. We have uploaded our structures to ModelArchive - <https://www.modelarchive.org/doi/10.5452/ma-jd-viral>. ModelArchive hosts predicted structures in a uniform way with extensive metadata. Furthermore, ModelArchive is part of EBI's 3D-Beacons framework (<https://www.ebi.ac.uk/pdbe/pdbe-kb/3dbeacons/>), enabling uniform downloads and processing of our protein structures through a shared API encompassing the PDB, AlphaFold database, and other databases.
3. We are in the process of making our viral structures available through Viralzone (<https://viralzone.expasy.org/>), a heavily-used viral resource hosted by the Swiss Institute of Bioinformatics. Here, structures will be available through a central landing page as well as through pages for each individual viral genus.

2) The authors present several new interesting hypotheses about the function of viral protein clusters. In several cases these observations already exist in the literature (poxvirus gasdermin and galactosyltransferase homologs) limiting demonstration of the impact of the authors' new large-scale structural database. Can the authors highlight further examples of connections from their database or provide further experimental data to test the novel hypotheses related to viral transporter-like or dioxygenase-like protein function?

Response: We have made several additional connections, and we provide more bioinformatic support for our finding of similarity between a family of herpesvirus proteins and eukaryotic nucleoside transporters.

1. Virus-encoded DNA-binding proteins

The potential biotechnology applications of DNA-binding proteins (such as in genome editing and diagnostics) spurred us to investigate new DNA binding proteins.

Many viruses manipulate host TATA-binding proteins to enhance viral replication or repair. We found that virus-encoded TATA-binding proteins are widespread amongst large dsDNA viruses, and we double the number of viral families known to encode these proteins. This suggests that many large dsDNA viruses use virus-encoded TATA-binding proteins to promote viral replication.

Fig. 2G:

G TATA DNA-Binding Protein (Cluster 215)

In addition, many poxviruses encode the I3L family of ssDNA-binding proteins. Notably, the structural basis of this activity has been enigmatic. We find that I3L is representative of a widespread family of ssDNA-binding proteins present across dsDNA viruses (See **Fig. 2H**, below). I3L contains an oligonucleotide-binding fold (OB-fold) that is similar to baculovirus DNA-binding protein 1 (DBP-1) and phage T7 single stranded binding protein (SSB), underscoring their shared ssDNA-binding behavior. Notably, we find that these eukaryotic virus I3L-like proteins contain a distinctive and conserved N-terminal beta sheet (See **Fig. 2I**, below) that is absent in other OB-fold containing viral proteins such as Baculovirus LEF-3 or Phage T7 SSB.

Fig. 2H:

H

Fig. 2I:

We discuss these findings in more detail on page 7 of the revised text.

2. Herpesvirus nucleoside transporters

We provide two additional lines of support for our hypothesis that UL43-like herpesvirus proteins are likely to be nucleoside transporters.

i) We use DALI to conduct structural alignments between Epstein-Barr virus BMRF2, a UL43-like protein, and structures categorized by the Transporter Classification Database. This analysis reveals that BMRF2 is structurally similar to human ENTs 1-4, and maintains some similarity to other families of transporters such as the peptide transporter (POT/PTR) family, Ferroportin (Fpn) family, and Major Facilitator Superfamily (MFS) which transport diverse substrates.

Fig. 3E:

ii) Phylogenetic analysis of Herpesvirus and related mammalian nucleoside transporters revealed that UL43-like proteins are prevalent across diverse herpesviruses. Notably, we identify a variant encoded by *Felis Catus* Gammaherpesvirus that maintains 36% sequence similarity to human ENT1, suggesting a structural and perhaps functional connection between these herpesvirus proteins and ENTs.

Fig 3F:

We discuss these findings on page 9 of the revised text. Notably, we mention that while there is strong support for these proteins being similar to nucleoside transporters, the exact substrates of these proteins are unknown.

3) The authors' data defining avian poxvirus LigT-family phosphodiester enzymes as inhibitors of cGAS-STING immunity is incomplete to support claims in the paper.

3a. The claims "enzymatic activity validated in cell based experiments" and "a novel mechanism of 2'3' cGAMP degradation by eukaryotic viruses" are not fully proven by the data. What is the evidence that LigT-like proteins are enzymes? Could these proteins instead bind and sequester 2'3'-cGAMP or prevent STING activation through an indirect mechanism?

3b. It is confusing why experiments are not performed to directly test the ability of viral LigT-like proteins to degrade 2'3'-cGAMP and related nucleotide immune signals. The authors are world-leading experts in biochemistry and several readily available methods exist that could be used to measure the ability of LigT-like

proteins to degrade 2'3'-cGAMP. For example, the authors could test viral LigT-like proteins using recombinant protein and commercially available 2'3'-cGAMP in degradation analysis by TLC, HPLC, or ion-exchange chromatography. Demonstrating that viral LigT-like proteins use a similar mechanism to degrade 2'3'-cGAMP as phage Acb1 proteins would help verify the proposed structural connection.

Response: We performed this experiment as suggested, providing direct evidence for the enzymatic function of LigT-like proteins. We purified wildtype and mutant Pigeonpox PDE, as well as Phage T4 Acb1 and the 2'5' OA-targeting LigT-like PDE NS2a from MHV, and analyzed 2'3' and 3'3' cGAMP cleavage using thin-layer chromatography. Similarly to Acb1 and unlike NS2a, Pigeonpox PDE cleaves both 2'3' and 3'3' cGAMP. This activity is eliminated when the catalytic histidines are mutated.

Fig 4E:

Furthermore, we find that the degradation products of 2'3' and 3'3' cGAMP from Acb1 and Pigeonpox PDE co-migrate via TLC, suggesting that these enzymes use a conserved mechanism of cGAMP hydrolysis.

Extended Data Fig. 9C:

Minor comments:

4) The title of the paper includes “birth of new protein folds” but new protein folds are not described or verified. What is the evidence of new protein folds (compared to proteins that simply do not have robust matches in FoldSeek or DALI analysis)?

Response: We have revised the title to read “Birth of protein folds and functions in the virome” to emphasize the clear emergence of structures and associated functions within evolving viral genomes and to acknowledge that these cannot at present be proven to be new.

5) It is unclear what the authors mean by describing protein folds as “evolutionarily young”? Are all predicted structures that have rare matches necessarily recently evolved?

Response: Similarly to our previous response, we have clarified our language. We now refer to these folds as “structurally distinct”.

6) In structural alignments between annotated and unannotated sequence clusters (Figure 2), can the authors additionally show the structure of the unannotated sequence colored by LDDT? Are high confidence residues driving the alignment score?

Response: We have moved the proteins in question to Extended Data Fig. 4, and now show the structures colored by pLDDT. Residues with high confidence levels, with pLDDTs generally above 70, anchor these alignments.

7) In the clustering analysis, 20% (>12,000) of the total viral sequences cluster as singletons (Figure S1B). Can the authors comment further on what this means? Is this due to limited MSAs, low confidence structure predictions, or possible correlations with multi-domain proteins or disordered regions?

Response: Because the pLDDTs of singletons are substantially lower than the pLDDTs of non-singletons, we suspect that the low prediction quality of singletons is the major driver of the inability to identify structural similarities. We attempted to improve prediction quality by generating MSAs against a larger reference database (our structures use MSAs generated exclusively against viral sequences). This analysis revealed that singletons have lower MSA depth that correlates with their lower pLDDTs, and that changing the MSA generation method has little impact. We have added the following information to the text:

Extended Data Fig. 3:

Page 4:

“Proteins in single-member clusters have substantially lower pLDDTs than those in non-singleton clusters (Extended Data Fig. 3A), suggesting that structure prediction quality has a major impact on our ability to detect structural similarity. We tested if MSA generation against a larger reference database has an impact on prediction quality. We found that while singletons have a lower average MSA depth that correlates with their lower pLDDT, this alternative MSA generation led to negligible effect on structure prediction quality (Extended Data Fig. 3B-D).”

8) In Supplemental Figure 2 It is unclear what the numbers beneath each “wheel” mean. Is this simply a cluster ID?

Response: Correct. We have added this information to the figure legend.

9) Description of poxin 2'3'-cGAMP nucleases should cite the original primary work of Eaglesham et al 2019 (PMID 30728498).

Response: We have corrected this omission.

I hope the authors will find my comments useful, thank you for the opportunity to read this interesting manuscript.

Philip Kranzusch (with Sam Fernandez and Sam Hobbs)

Referee #1 (Remarks on code availability):

The database and code do not appear to be available on Zenodo. This is a major point in me review, the accessibility of these data is a key part of the impact of the research.

Response: As mentioned above, the structures are now available through multiple sources:

- Zenodo: <https://zenodo.org/doi/10.5281/zenodo.10291580>
- Google colab: <https://colab.research.google.com/github/jnoms/vpSAT/blob/main/bin/colab/ExploreStructures.ipynb>
- ModelArchive: <https://www.modelarchive.org/doi/10.5452/ma-jd-viral>

The code is freely available on Github:

- SAT: <https://github.com/jnoms/SAT>
- vpSAT: <https://github.com/jnoms/vpSAT>

The code is also available on Zenodo. This Zenodo repository also contains all intermediate files necessary to reproduce the figures:

<https://zenodo.org/doi/10.5281/zenodo.10373132>

Referee #2 (Remarks to the Author):

The manuscript by Nomburg et al. describes an extensive sequence and structural analysis of the proteomes of 4,463 viral species across 132 different viral families. The clustering analysis recapitulates many evolutionary tenets: proteins central to the viral cycle are predominant, structure conservation is far more extent than sequence conservation. One interesting point is the finding of many clusters with significant structural homology with clusters of other viral families, as well as

non-viral clusters, suggesting the presence of horizontal gene transfer. The most important contribution of the paper appears the existence of two acquisition events within the betacoronavirus genus, of PDEs, proteins involved in the evasion of the oligoadenylate synthase pathway, a highly conserved immune mechanism against double-stranded viruses.

In my opinion, the manuscript is rigorous, polished and technically excellent, it is well-written and it addresses interesting questions in fundamental evolutionary biology. The analysis is clear and intuitive, and the scale of the computational work is impressive. In my opinion, the article should definitely be published in some form, and I would like to congratulate the authors for their excellent work.

Response: Thank you for the positive feedback.

The main conclusion appears to be the discovery of the genetic signatures of the formation of the pathway. While I am not a virologist and therefore cannot fully comment on the relevance of this finding, it does not appear sufficiently novel to warrant publication. From the field of structural bioinformatics — which is my field of expertise, and why I suspect I was asked to review this paper —, although the authors' methodology is sound and the scale of application is impressive, I do not find their methods particularly novel. Given that the authors also only seem to find one new piece of information, I am not convinced that the methodology provides a general method for hypothesis generation and/or testing to characterise viral protein evolution and function across the virome.

I also have a few minor comments to provide to the authors:

- The authors generate protein structure predictions using ColabFold, although with manually generated MSAs to enhance performance. The alignments are generated using MMseqs2 on the RefSeq virus database. While I understand the authors' choice, some publications, if not popular folklore around AlphaFold usage, suggests that incorporating metagenomics sequences into the alignment is crucial for getting good predictions in rapidly evolving proteins. I was wondering if the authors would be able to provide an empirical validation of the effect of incorporating all databases to the alignment (i.e. UniRef + BFD + Mgnify), even if only for a few representative proteins. This test would have to be done not only on the larger clusters (where there are obviously many homologs) but also on some of the ~60% clusters that have a single representative.

Response: To address this comment, we took the protein cluster representatives from the 100 largest clusters and from 100 random singleton clusters. We generated MSAs through the Colabfold MMseqs2 server which, according to the Colabfold methods, conducts an alignment against data within UniRef, BFD and Mgnify. We found that this new MSA generation strategy had a negligible impact on prediction quality (or even a very

slight negative effect on the prediction quality of singletons). Notably, singletons have a much lower prediction quality than non-singletons (and an associated lower MSA depth).

We've included the following in the text:

Extended Data Fig. 3:

Page 4:

“Proteins in single-member clusters have substantially lower pLDDTs than those in non-singleton clusters (Extended Data Fig. 3A), suggesting that structure prediction quality has a major impact on our ability to detect structural similarity. We tested if MSA generation against a larger reference database has an impact on prediction quality. We found that while singletons have a lower average MSA depth that correlates with their lower pLDDT, this alternative MSA generation led to negligible effects on structure prediction quality (Extended Data Fig. 3B-D).”

- The authors perform clustering at the sequence level and then use different structural techniques to understand shared evolutionary relations, which is illuminating. Nevertheless, there is one test that I was missing: if one runs structural searches for the ~12,000 clusters that have only one representative against the entire ~60k structures, is it possible to find structural homologs that were previously ignored? I would first test this using Foldseek, although intuition suggests (because the structure predictions may be of lower quality for these isolated clusters due to the lack of homologs) that DALI's increased sensitivity will be necessary for this experiment.

Response: We addressed this point in two ways:

1. As suggested, we used Foldseek to align each singleton with the entire structure database. This revealed that **224** of the 12,422 singletons align to another protein with a TMScore of at least 0.4 and alignment coverage of at least 70%
2. The slow speed of DALI searches prevents us from doing a search similar to what we did with Foldseek. Instead, we conducted DALI alignments between each of the 12,422 singletons and the protein cluster representative from all 5,770 non-singleton clusters. This revealed that **200** singletons align to non-singleton cluster representatives with a Z score of at least 8 and alignment coverage of at least 70%. Of these 200, 10 also are part of the 224 that were aligned above with foldseek.

Ultimately, we find that the vast majority of singletons are structurally distinct from other proteins in our dataset. As we show above, this is likely a consequence of poor prediction quality.

In accordance with my comments above, my recommendation to the editor would be that the manuscript is transferred to another journal of the Nature family (e.g. Nature Communications) without further review, where my recommendation would be of acceptance after addressing the minor comments posed above.

Referee #3 (Remarks to the Author):

Romburg et al used modern bioinformatic tools such as Colabfold to predict the structures of more than 60,000 proteins of eukaryotic viruses that have been sequenced. They found that 62% of these viral proteins do not have homologs in the AlphaFold database. For those viral proteins that have homologs in the AlphaFold database, putative functions of these viral proteins can be inferred based on homology to proteins with known functions. The authors focused on a class of proteins that contain the RNA ligase T (LigT)-like phosphodiesterase (PDE) domain, which has been found in the bacteriophage protein Acb1, an enzyme known to degrade the second messenger molecule cGAMP. They tested some of these putative enzymes and found that the LigT-like PDEs from pigeon and canary could indeed specifically inhibit STING activation by 2'3'-cGAMP.

The structural prediction of eukaryotic viral proteins provides a very useful resource for the community to study the functions of many viral proteins including those that have not been annotated. The authors validated their approach by providing an example of a class of viral enzymes that degrade cGAMP. However, the characterization of these enzymes is quite preliminary, especially considering that another class of cGAMP PDE has already been reported in poxviruses (known as poxin; PMID 30728498). This reviewer suggests that the authors further characterize the LigT family of enzymes as outlined below:

1) The reporter assay shown in Figure 4E does not prove that the avian poxvirus proteins cleave cGAMP; the results only showed that the proteins inhibited STING activation by cGAMP. In fact, several of the viral proteins (WT and/or mutant) also inhibited STING activation by diABZI, which raises the possibility that some of these proteins might interfere with STING function through other mechanisms. The authors should carry out a biochemical assay to test directly whether the viral proteins indeed degrade cGAMP in vitro and if so, what are the degradation products. In Figure 4E and Figure S5A, they should also show Western blots to evaluate the protein expression levels in order to draw a valid conclusion.

Response: We have re-cloned our panel of PDEs with strep tags, and have conducted Western blots to measure expression. This analysis revealed that a subset of the PDEs express well in our system, whereas the PDEs from MHV (both wildtype and mutant), *Marseilleviridae*, Rotavirus and Pithoviridae show little observable expression (see Extended Data Fig. 9A, below). Given this observation, subsequent experiments focused on PDEs with observable expression. This leaves open the possibility that PDEs we are unable to test could have activity against cGAMP.

Extended Data Fig. 9A:

As suggested, we conducted a biochemical assay to test whether LigT-like PDEs encoded by an avian poxvirus (in this case, Pigeonpox) degrade cGAMP in vitro and to determine the degradation products. We purified wildtype and mutant Pigeonpox PDE, as well as Phage T4 Acb1 and the 2'5' OA-targeting LigT-like PDE NS2a from MHV, and visualized 2'3' and 3'3' cGAMP cleavage with thin-layer chromatography. Similarly to Acb1 and unlike NS2a, Pigeonpox PDE cleaves both 2'3' and 3'3' cGAMP. This activity is eliminated when the catalytic histidines are mutated.

Fig 4E:

m
Furthermore, we find that the degradation products of 2'3' and 3'3' cGAMP from Acb1 and Pigeonpox PDE co-migrate via TLC, suggesting that these enzymes generate conserved products and use a conserved mechanism of cGAMP hydrolysis. The major product of Acb1 degradation of 3'3' cGAMP has been reported as GpAp (PMID: 35395152).

Extended Data Fig. 9C:

2) The authors claimed that several of the ligT-like proteins from RNA viruses could cleave 2-5A but did not show any evidence. They should do the experiments to test the activity of these enzymes against 2-5A or other nucleotide messengers.

Response: There is extensive evidence in the literature that RNA virus-encoded LigT-like PDEs degrade 2'5' OA. We have added the following supporting references to the text: PMIDs: 22704621, 25758703, 28003490, 23878220, 32051268, 25878106.

3) Figure S5A shows that the ligT-like protein from MERS (NS4b) could also inhibit STING activation by cGAMP but did not do the control experiments such as diABZI stimulation or using a catalytic mutant. Considering that MERS is a human pathogen, performing these experiments can enhance the paper. Do SARS viruses also encode such ligT-like enzymes?

Response: We have added data to Extended Data Fig. 9B showing that MERS NS4b inhibits both 2'3' cGAMP- and diABZI-driven STING activity. This is consistent with multiple studies that show that MERS NS4b inhibits innate immune signaling independently of its antagonism of 2'5' OA (see PMIDs 24443473, 35338144, 26631542). Furthermore, biochemical analysis shows that MERS NS4b does not degrade 2'3' cGAMP (PMID 34372695). SARS-CoV-1 and SARS-CoV-2 do not contain LigT-like enzymes.

Extended Data Fig. 9B:

4) The authors should try to characterize several ligT-like proteins shown in Figure 4B for their enzymatic activity and substrate specificity (e.g, specificity to various cyclic dinucleotides). This is important for establishing this family of proteins as viral PDEs that antagonize host immune defense.

Response: We tested the activity of the Pigeonpox LigT-like PDE against cGAMP isomers in both our STING luciferase assay and in biochemical assays as described above.

In the STING luciferase assay, we find that the Pigeonpox LigT-like PDE, similarly to Phage T4 Acb1, prevents STING activation by 2'3', 3'3', and 3'2' cGAMP, but not by the non-nucleotide STING agonist diABZI. We confirmed that mutation of the catalytic histidines eliminates this activity. These results show that Pigeonpox PDE has broad cGAMP targeting specificity.

Figure 4D

D
As discussed above, we also show biochemical evidence that Pigeonpox PDE cleaves 2'3' and 3'3' cGAMP.

Altogether, these data show that Pigeonpox PDE has broad targeting specificity for cGAMP isomers.

Referee #4 (Remarks to the Author):

The manuscript by Nomburg, Price and Doudna describes comprehensive modeling a comparative structural analysis of viral proteins using AlphaFold2, FoldSeek and DALI. In addition, the authors experimentally test one of the resulting predictions, cGAMP like activity of viral PDEs that share the LigT fold.

AlphaFold2 is a breakthrough in protein structure and evolution analysis, but predictions of viral protein structures are conspicuously missing in the currently available database of AlphaFold predictions. The authors of this paper fill this gap and thus create a useful resource that will be appreciated and utilized by many researchers.

However, in the view of this reviewer, this is all that is achieved in this work, at least, as far as the computational part is concerned. The authors repeatedly claim novelty in terms of prediction of protein structures, evolutionary relationships and functions. However, the actual extent of novelty is impossible to assess, above all, because there is no effort to compare the performance of the structural analysis with that of sequence based methods, but also because some of the most relevant recent publications are neglected.

Response: There is extensive literature since the 1990s showing the benefit of structural comparison over sequence comparison (for example, PMIDs: 8566533, 7937731, 8771179). Nonetheless, we have now comprehensively benchmarked our structure-based analysis compared to sequence-based methods. These benchmarks are of two types: 1) Virus-virus comparisons, and 2) virus-nonvirus comparisons. These analyses confirmed that our approach provides a substantial improvement over sequence-based methods.

Our sequence + structure approach outperforms sequence-based methods for virus-virus protein comparisons.

In our manuscript, we use sequence and structure to identify viral protein clusters. We investigated the ability of three different sequence-based methods to reconstitute our viral protein clusters: 1) MMseqs2, 2) DIAMOND blastp, and 3) jackhmmer. Jackhmmer in particular is a highly sensitive HMM-based iterative search method.

We conducted the following experiment: For all protein clusters that contain at least two sequence clusters, we conducted all-by-all sequence alignments between all cluster members using the three sequence-based methods. Then, we performed connected-component clustering, and measured the number of clusters detected and the percentage of proteins that belong to the largest cluster generated through these sequence methods. These steps are illustrated in Extended Data Fig. 8A:

This analysis revealed that sequence-based methods were substantially less sensitive than our approach at identifying similar viral proteins. First, we find that sequence-based methods regularly fail to group all cluster members into a single cluster (see Extended Data Fig. 8B, below). jackhmmer has the best performance, but on average detects over

two distinct clusters. Sequence methods frequently generate a largest cluster that fails to incorporate all sequences (see Extended Data Fig. 8C, below).

Our structural comparisons outperform sequence methods at identifying similarities between viral and non-viral proteins.

In Figure 3, we highlighted four cases of structural similarity between viral and non-viral proteins. To do this, we conducted structural comparisons with DALI using one of four non-viral proteins as a query, and the viral structures as the target database. Here, we tested whether sequence-based methods are able to reconstitute hits identified by these DALI searches. A diagram explaining this procedure is shown at the top of Extended Data Fig. 8D, below. We find that jackhmmer, DIAMOND and MMseqs2 almost always fail to identify hits found by DALI.

Secondly, we conducted searches with hhPred, using specified viral proteins as queries. hhPred is a highly sensitive search method. Indeed, as mentioned above, hhPred is the method through which the poxvirus gasdermin-like structure was identified without using predicted structure (PMID 34253028). We confirm this finding, but show that viral proteins that are similar to COLGALT1, ENT4, and that contain a dioxygenase-like fold, do not yield informative hits through hhPred. See Extended Data Fig.8 E, below.

Extended Data Fig. 8D,E:

Altogether, we show that our structure-based methods outperform sequence-based methods at identifying virus-virus and virus-nonvirus protein similarities.

A characteristic example is the "discovery" of the poxvirus gasdermin homologs. The authors identify this relationship by structural comparison and cite a preprint by Elde and colleagues reporting the same. Incidentally, citing the preprint is misleading because a substantially updated version of the cited work has since been published:

<https://pubmed.ncbi.nlm.nih.gov/37494187/>

More importantly, however, minimal attention to the published literature would have revealed that the same discovery has been reported based solely on sequence comparisons, and the evolutionary history of gasdermins in poxviruses has been traced in detail: <https://pubmed.ncbi.nlm.nih.gov/34253028/>.

Response: We have updated our references accordingly.

This example shows that without systematic benchmarking, the impact of structural analysis remains uncertain.

Response: As discussed above, comprehensive benchmarking establishes that structural analysis increases sensitivity over sequence-based methods.

Even in cases where structural comparison is indeed important for discovering relationships between proteins, the authors overlook preceding publications that

happen to report a far deeper analysis. A case in point is the poxvirus C1 protein for which the authors report the dioxygenase fold. In a previous publication, AlphaFold2/DALI were employed to identify this fold not only in C1 but in several other poxvirus proteins, and importantly, the patterns of inactivation of the catalytic sites of this enzyme were explored:

<https://pubmed.ncbi.nlm.nih.gov/37017580/>. This case is particularly suspicious because Nomburg et al only report one poxvirus dioxygenase homolog.

Response: We only highlighted a single example, but observe many (if not all) cases discussed in the cited manuscript.

Even when it comes to the "crown jewel" of this work, the LigT-like PDEs, the degree of novelty is unclear given this recent publication:

<https://pubmed.ncbi.nlm.nih.gov/38277437/>

(also cited as a preprint although in this case, article was not yet published at submission of Nomburg et al.)

Response: The manuscript cited above is tangential to our main findings, and has no impact on the novelty of our results. The cited manuscript assesses the phylogenetic history of RNA virus-encoded LigT-like PDEs. However, it does not consider the connection between RNA virus LigTs and cGAMP-degrading enzymes encoded by phage, nor does it assess LigT-like PDEs that are encoded by eukaryotic DNA viruses.

In summary, without careful benchmarking and incorporation (and proper citation) of previously published work, the analysis presented in this manuscript has virtually no merit.

The manuscript by Nomburg et al often refers to evolutionary events such as horizontal gene transfer, but the analysis leading to conclusions on such events is rudimentary and methodologically faulty. For instance, the authors use BLAST scores to infer horizontal gene transfer which is strict no go in evolutionary biology. Hand in hand with this methodological inadequacy comes inappropriate language such "sequence homology" and structural homology" that appear throughout the manuscript.

Response: Thank you for this comment. We have completely modified our phylogenetics workflow. We now identify similar sequences using PSI-Blast against the NCBI nr database, and construct phylogenetic trees using Fasttree (see the methods section for more details). This augmented analysis supports our initial results. In addition, we have removed usage of the word "homology" in favor of "similarity".

Reviewer Reports on the First Revision:

Referees' comments:

Referee #1 (Remarks to the Author):

The revised manuscript by Nomburg et al is significantly improved. I recommend publication and commend the authors on an exciting study.

Minor comments:

- 1) Consider adding to the Colab notebooks the option to generate Foldseek search results as HTMLs, a functionality that already exists as an additional flag in Foldseek search calls (as “- - format-mode 3”) to allow users to interactively visualize alignment results.
- 2) Consider adding a colorbar to the figures to indicate the range of pLDDT scores.
- 3) Consider defining “MSA depth” in the figure legend or method section.

Philip Kranzusch (with Sam Fernandez and Sam Hobbs)

Referee #3 (Remarks to the Author):

This revision has addressed my concerns, which were focused on the biochemistry of the viral PDEs the authors identified from their bioinformatic analyses. For structural bioinformatics which constitutes the major part of this paper, I defer to experts on this subject matter.

Referee #4 (Remarks to the Author):

In the revised manuscript, the authors include an enormous amount of additional work that certainly makes the manuscript more informative and thorough.

Nevertheless, some issues remain. The benchmarking of sequence-based vs structure-based approaches for detecting remotely similar homologs of viral proteins is rather perfunctory. The comparison of clustering tells little about the actual relative power of these approaches. Then, this is followed by an anecdotal assessment of several novel findings. Certainly, better than nothing but, in the opinion of this reviewer, a straightforward quantitative comparison is what is needed. For example, for how many 'orphan' viral proteins cellular homologs could be detected with significant scores by sequence vs structure comparison.

More on the technical side, I strongly suggest using HHPred for sequence comparison. Jackhammer is a good method, but HHPred, in general, performs even better.

Ditto for inferences of HGT. In the revision, a phylogenetic approach is used instead of BLAST hits

comparison (which as completely inadequate in the original manuscript). However, FastTree is a crude method and not state of the art. I strongly suggest using the latest version of IQTree.

Inexplicably, a key publication reporting some of the major findings reported here <https://pubmed.ncbi.nlm.nih.gov/37017580/> is still not cited although mentioned in the rebuttal letter.

Author Rebuttals to First Revision:

Referee #1 (Remarks to the Author):

The revised manuscript by Nomburg et al is significantly improved. I recommend publication and commend the authors on an exciting study.

Response: Thank you for your positive feedback.

Minor comments:

1) Consider adding to the Colab notebooks the option to generate Foldseek search results as HTMLs, a functionality that already exists as an additional flag in Foldseek search calls (as “- -format-mode 3”) to allow users to interactively visualize alignment results.

Response: Done.

2) Consider adding a colorbar to the figures to indicate the range of pLDDT scores.

Response: Done. We have added pLDDT colorbars to figures containing structures that are colored by pLDDT.

3) Consider defining “MSA depth” in the figure legend or method section.

Response: Done. We added this to the Extended Data Fig 3B figure legend.

Philip Kranzusch (with Sam Fernandez and Sam Hobbs)

Referee #3 (Remarks to the Author):

This revision has addressed my concerns, which were focused on the biochemistry of the viral PDEs the authors identified from their bioinformatic analyses. For structural bioinformatics which constitutes the major part of this paper, I defer to experts on this subject matter.

Response: Thank you for your positive feedback.

Referee #4 (Remarks to the Author):

In the revised manuscript, the authors include an enormous amount of additional work that certainly makes the manuscript more informative and thorough.

Nevertheless, some issues remain. The benchmarking of sequence-based vs structure-based approaches for detecting remotely similar homologs of viral proteins is rather perfunctory. The comparison of clustering tells little about the actual relative power of these approaches. Then, this is followed by an anecdotal assessment of several novel findings. Certainly, better than nothing but, in the opinion of this reviewer, a straightforward quantitative comparison is what is needed. For example, for how many 'orphan' viral proteins cellular homologs could be detected with significant scores by sequence vs structure comparison.

More on the technical side, I strongly suggest using HHPred for sequence comparison.

Jackhammer is a good method, but HHPred, in general, performs even better.

Response: Thank you for this helpful feedback. We have conducted a quantitative comparison of the ability of structural and sequence searches to identify cellular homologs of “orphan” viral proteins.

We established a set of proteins for this comparison as follows:

- We identified sequence clusters that contained more than 1 member, and for whom less than ¼ of members had an Interproscan alignment. This resulted in 4,409 sequence clusters. We used a representative from each cluster for further analysis.
- From this set, we selected sequence cluster representatives that were well folded, with an average pLDDT of at least 70. This resulted in a final set of 1,326 proteins.

We then aligned these proteins against the PDB, using DALI for structural search and HHpred for sequence search, as follows:

- Because of the substantial runtime of DALI searches, we aligned each viral protein against the sequence clustered “PDB25” structure database provided by the DALI authors. We considered alignments with a Z score of at least 7 to be significant alignments.
- HHpred is a web server not amenable to thousands of searches. Thus, we reconstituted HHpred locally. First, we used HHblits to iteratively align each viral protein against the HHSuite-provided Uniref30 database. Next, we used MSAs generated from these alignments as queries for HHsearch against the HHSuite-provided PDB database. We considered alignments with an E-value less than 0.001 to be significant alignments.

Of the 1,326 proteins, DALI found significant alignments for 661 proteins, compared to 295 proteins by hhPred. These data indicate that structural alignments with DALI are more sensitive than HHpred-based sequence searches for proteins with high-quality structure predictions.

We have added these data as Extended Data Fig 8E and 8F, which we show below:

Ditto for inferences of HGT. In the revision, a phylogenetic approach is used instead of BLAST hits comparison (which as completely inadequate in the original manuscript). However, FastTree is a crude method and not state of the art. I strongly suggest using the latest version of IQTree.

Response: We have now modified our phylogenetic approach, and use IQTree as suggested.

Inexplicably, a key publication reporting some of the major findings reported here <https://pubmed.ncbi.nlm.nih.gov/37017580/> is still not cited although mentioned in the rebuttal letter.

Response: This publication was cited in our prior submission (see reference 51 in our prior submission).

We have added text to give additional context to this citation (reference 47 in this version). The added text is in bold.

Page 8:

“

In addition, we observed structural similarity of Poxvirus C4-like proteins with eukaryotic dioxygenases (Extended Data Fig. 5C), consistent with previous work that identified frequent exaptation of inactivated host enzymes by poxviruses⁴⁷.

“

Reviewer Reports on the Second Revision:

Referees' comments:

Referee #4 (Remarks to the Author):

In this revised manuscript, the authors have addressed my concerns.